# Cancer cell genetics shaping of the tumor microenvironment reveals myeloid cell-centric exploitable vulnerabilities in hepatocellular carcinoma

Christel F. A. Ramirez[1,2,6], Daniel Taranto[1,2,6], Masami Ando-Kuri[1,2,6], Marnix H. P. de Groot[1], Efi Tsouri[1,2], Zhijie Huang[3], Daniel de Groot[1], Roelof J. C. Kluin[4], Daan J. Kloosterman [1,2], Joanne Verheij[5], Jing Xu [1,2,3], Serena Vegna [1,2,7] ✉ & Leila Akkari [1,2,7] ✉

Myeloid cells are abundant and plastic immune cell subsets in the liver, to which pro-tumorigenic, inflammatory and immunosuppressive roles have been assigned in the course of tumorigenesis. Yet several aspects underlying their dynamic alterations in hepatocellular carcinoma (HCC) progression remain elusive, including the impact of distinct genetic mutations in shaping a cancer-permissive tumor microenvironment (TME). Here, in newly generated, clinically-relevant somatic female HCC mouse models, we identify cancer genetics' specific and stage-dependent alterations of the liver TME associated with distinct histopathological and malignant HCC features. Mitogen-activated protein kinase (MAPK)-activated, $Nras^{G12D}$-driven tumors exhibit a mixed phenotype of prominent inflammation and immunosuppression in a T cell-excluded TME. Mechanistically, we report a $Nras^{G12D}$ cancer cell-driven, MEK-ERK1/2-SP1-dependent GM-CSF secretion enabling the accumulation of immunosuppressive and proinflammatory monocyte-derived $Ly6C^{low}$ cells. GM-CSF blockade curbs the accumulation of these cells, reduces inflammation, induces cancer cell death and prolongs animal survival. Furthermore, GM-CSF neutralization synergizes with a vascular endothelial growth factor (VEGF) inhibitor to restrain HCC outgrowth. These findings underscore the profound alterations of the myeloid TME consequential to MAPK pathway activation intensity and the potential of GM-CSF inhibition as a myeloid-centric therapy tailored to subsets of HCC patients.

Hepatocellular carcinoma, the most common form of primary liver cancer, is a highly heterogenous disease both at the pathological and molecular levels[1–3]. These tumors exhibit high intra- and inter-tumor heterogeneity, which challenges the development of effective therapies for advanced HCC patients[4]. As a consequence, HCC is a leading cause of cancer death worldwide[5]. Different environmental risk factors, such as viral hepatitis, metabolic syndromes or alcohol abuse contribute to the multifaceted HCC molecular pathogenesis[4], fueled by the underlying chronic inflammatory background characterizing all etiologies[6].

While pivotal to define precise molecular and immune HCC subclasses[4], extensive genetic, epigenetic and transcriptomic analyses over the last decade have failed to advance novel precision medicine treatments targeting liver cancer cells[4]. Meanwhile, immune and stromal cell targeting strategies based on VEGF inhibition and program death-ligand 1 (PD-L1) blockade have shown unprecedented results[7–9], being the first treatment regimen with improved overall survival relative to Sorafenib, a decade-long mainstay treatment for advanced-stage HCC patients[10,11]. However, the multifaceted heterogeneity of the TME remains a major challenge to immunotherapy efficacy, a therapeutic approach only benefitting a minority of HCC patients[12,13].

Mounting evidence suggests that different cancer cell-intrinsic features, such as genetic make-up and/or signaling pathway deregulation, play a critical role in shaping different TMEs[14,15], thus emphasizing the need to implement tailored immunomodulation strategies according to cancer molecular profiles. In HCC and other solid tumor types, well-established oncogenic signals such as Myc[16–19], Wnt/β-Catenin[12,20,21], Ras-MAPK-ERK pathway activation or loss of the tumor suppressor gene *Tp53*[22–26] distinctively reprogram the liver local and systemic environment through either promoting inflammation, immunosuppression or dampening anti-tumor immunity[27]. Importantly, quantitative differences in the Ras-MAPK-ERK signaling pathway activation state can exert unique biological consequences in response to extrinsic cues specific to different TMEs[28]. Yet, the non-cell autonomous effects and cellular mediators ensuing oncogenic pathways activation threshold that may dictate the HCC TME landscape remain to be identified.

It is well established that a dynamic interplay between cancer cells and the immune system impacts disease progression and response to therapy[15,21,29–34], a process further complexified when considering the fine balance between immunotolerance and inflammatory wound healing response peculiar to the liver regenerative capacity. Myeloid cells constitute a heterogenous and dynamic population of functionally plastic innate immune cells, whose recruitment, differentiation and activation are influenced by cancer- and stromal-derived cues[6,35,36]. Reciprocally, tumor-educated myeloid cells promote cancer cell proliferation, immunosuppression and immune evasion[6,35,37–39]. For instance, macrophage and NK cell recruitment and pro-tumorigenic functions are influenced by p53 expression in hepatic stellate cells or transformed hepatocytes[25,40], respectively. Moreover, such effects are reversible, as exemplified by p53 restoration in cancer cells, which curbed the myeloid-driven immunosuppressive TME and reestablished anti-tumor immune surveillance[41]. Hence, to apply personalized immunomodulation strategies overcoming the limited effects of T cell-centric immunotherapy, the mechanisms underlying HCC inter-patient heterogeneity that impact myeloid cell subset composition and activation ought to be unraveled.

In this work, we present a collection of preclinical murine models of HCC mimicking the deregulation of oncogenic signaling pathways commonly altered in HCC patients[26,42–46] and recapitulating distinct molecular and histopathological features characteristic of human HCC subclasses. Utilizing complementary multi-omics approaches and functional assays, we reveal the heterogeneity of the myeloid cell pool and identify dichotomic proinflammatory and immunosuppressive features fueling HCC pathogenesis in tumors presenting heightened MAPK activity. Mechanistically, a cancer cell-intrinsic MEK-ERK1/2-SP1 signaling cascade enforces expression and secretion of GM-CSF, an exploitable vulnerability synergistic with the current standard of care treatment for HCC employing VEGF neutralization. Overall, our results highlight the importance of inter-patient heterogeneity and stratification, unveiling distinctive combinatorial immunomodulatory strategies as a potential therapeutic avenue for HCC patients.

## Results

### Genetically-distinct liver cancer mouse models display unique histopathological and molecular features

To address the intricate relationships between distinct oncogenic signaling pathway activation and immune cell shaping in hepatocarcinogenesis, we generated genetically-distinct, immunocompetent HCC mouse models using hydrodynamic tail vein injection (HDTVi)-delivery of genetic elements[47–49]. These preclinical models comprised combinations of oncogene overexpression and tumor suppressor gene knockout mimicking frequently altered signaling pathways in HCC patients (Fig. 1a). *Myc*, amplified in up to 30% of HCC patients[26,43], was the oncogenic driver used in two of these HCC preclinical models, in combination with either loss of *Trp53* (mutated in up to 50% of HCCs[44]) or activation of AKT/PI3K/mTOR signaling pathway (activated in 50% of HCC patients[45]) through *Pten* deletion. Furthermore, we mimicked the activation of the MAPK signaling pathway (aberrantly activated in ~50% of HCCs[46]) using constitutive expression of the proto-oncogene *Ras*. In light of the evident differences in MAPK activation states and cancer malignant features incumbered by distinct Ras isoforms and point mutations in several cancers[50–54], including HCCs[28,41,55], we employed distinct oncogenic mutations of *Nras*: *Nras*^G12D (*pT3-Nras*^G12D-GFP) and *Nras*^G12V (*pT/CaggsNras*^G12V-IRES-Luc), to generate two additional HCC models, both combined with *Pten* loss.

We first confirmed the oncogenic driver-mediated activation/inhibition of these pathways in established *Myc*^OE/*Trp53*^KO, *Myc*^OE/*Pten*^KO, *Nras*^G12D/*Pten*^KO, and *Nras*^G12V/*Pten*^KO liver tumors (Supplementary Fig. 1a). Tumor development and disease progression were assessed in each model using longitudinal magnetic resonance imaging (MRI; Supplementary Fig. 1b) and survival curves were generated for each models following HDTVi-mediated tumor induction, with varying tumor penetrance rates (Fig. 1b, c and Supplementary Fig. 1b). Both *Myc*^OE-driven liver cancer models displayed multinodular tumors with brisk tumor latency and rapid outgrowth, consequently leading to short median survivals. This effect was irrespective of the invalidated *Trp53* or *Pten* tumor suppressor genes, thus confirming the dominant role of MYC in cancer progression[26,56]. Contrastingly, HCCs driven by *Nras* overexpression combined with *Pten*^KO displayed longer median survival, with different *Nras* point mutations resulting in distinct tumor penetrances, growth rates and nodule numbers. Indeed, *Nras*^G12D point mutation gave rise to single-nodular tumors with a shorter latency compared to *Nras*^G12V multifocal HCC, which presented the longest median survival of all four liver cancer models. To exclude the use of distinct *Nras* vector backbones as a factor influencing these murine model features, we generated an additional *Nras*^G12V/*Pten*^KO HCC model by directly mutating the *pT3-Nras*^G12D-GFP vector in codon 12 from D to V, thus engineering a pT3-*Nras*^G12V/*Pten*^KO HCC mouse model. We verified that pT3-*Nras*^G12V/*Pten*^KO HCC model recapitulated the features of *pT/CaggsNras*^G12V-IRES-Luc *Nras*^G12V/*Pten*^KO (Supplementary Fig. 1c), confirming that the aggressivity of the distinct *Nras*-driven models was subsequent to the introduced *Nras* point mutations in hepatocytes.

We next investigated the histopathological and systemic malignant features of the genetically-distinct liver cancers models. We first confirmed the hepatocytic origin of these tumors with the readout of Arginase-1 expression[57] (Supplementary Fig. 1d, e). Increased systemic alpha fetoprotein (AFP)[58] (Supplementary Fig. 1f) and bulk tumor *Afp* expression (Supplementary Fig. 1g) validated all four models as bona fide HCCs. Additionally, accumulation of seric alanine amino transferase (ALT) activity was evident in *Myc*^OE/*Trp53*^KO, *Myc*^OE/*Pten*^KO, and *Nras*^G12V/*Pten*^KO tumor-bearing mice (Supplementary Fig. 1h), indicating increased liver damage in multinodular tumors[59]. Consistent with the role of PI3K/AKT activation in triggering features of fatty liver disease[60], lipid accumulation was observed in *Pten*^KO tumors. However, overt fibrosis and steatosis characterized *Nras*^G12D/*Pten*^KO and *Nras*^G12V/*Pten*^KO tumors, respectively (Fig. 1d and Supplementary Fig. 1i). Pathological assessment revealed that *Myc*^OE-driven tumors were

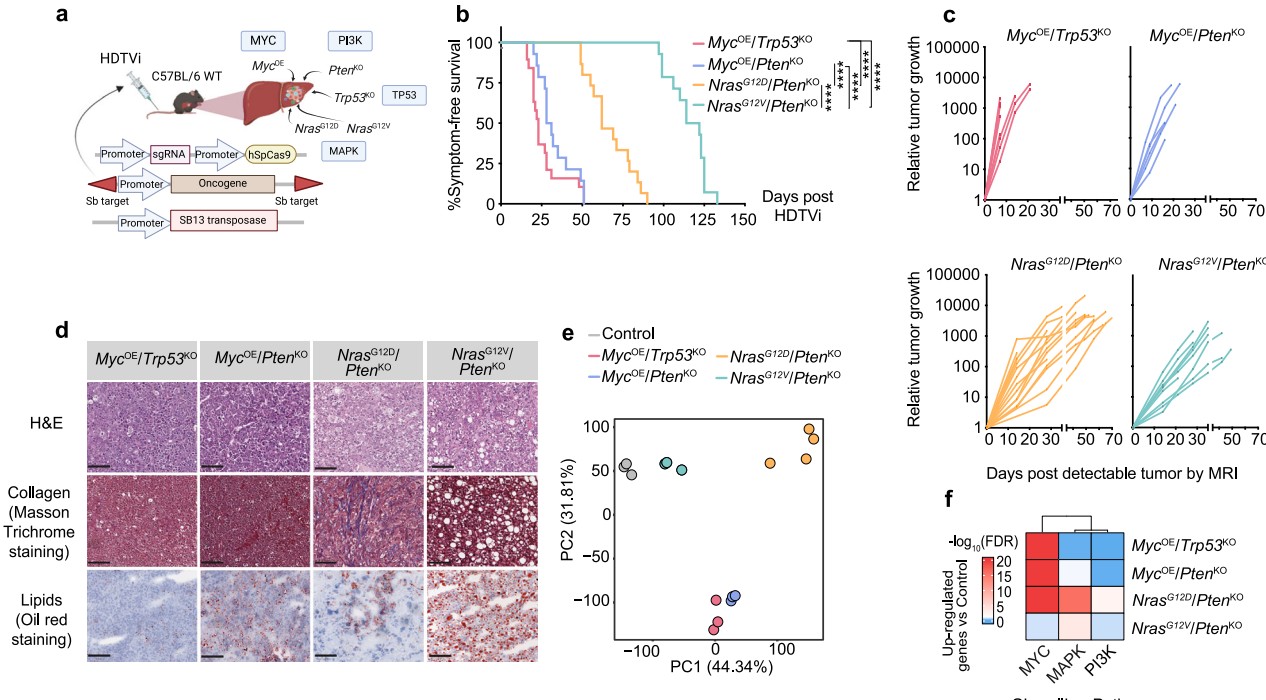

**Fig. 1 | Genetically-distinct murine HCCs display distinct transcriptomic and histopathological features. a** Schematic diagram of experimental design: Hydrodynamic tail vein injection (HDTVi) method used to deliver the Sleeping beauty (SB) transposon and CrispR-Cas9 constructs that enforce expression of the depicted oncogenic drivers in hepatocytes in order to generate genetically-distinct HCCs. **b** Kaplan–Meier survival curves of HCC-bearing mice HDTV injected with the indicated combinations of oncogenic drivers ($Myc^{OE}/Trp53^{KO}$ n = 19, median survival of 23 days; $Myc^{OE}/Pten^{KO}$ n = 14, median survival of 29.5 days; $Nras^{G12D}/Pten^{KO}$ n = 15, median survival of 62 days; $Nras^{G12V}/Pten^{KO}$ n = 14, median survival of 118 days). $Nras^{G12V}/Pten^{KO}$ vs $Nras^{G12D}/Pten^{KO}$ p < 0.001; $Nras^{G12V}/Pten^{KO}$ vs $Myc^{OE}/Trp53^{KO}$ p < 0.001; $Nras^{G12V}/Pten^{KO}$ vs $Myc^{OE}/Pten^{KO}$ p < 0.001; $Nras^{G12D}/Pten^{KO}$ vs $Myc^{OE}/Trp53^{KO}$ p < 0.001; $Nras^{G12D}/Pten^{KO}$ vs $Myc^{OE}/Pten^{KO}$ p < 0.001; $Myc^{OE}/Trp53^{KO}$ vs $Myc^{OE}/Pten^{KO}$ n.s. (non-significant). ****p < 0.0001 **c** Longitudinal HCC volumes of individual tumor-bearing mice ($Myc^{OE}/Trp53^{KO}$ n = 9, $Myc^{OE}/Pten^{KO}$ n = 6, $Nras^{G12D}/Pten^{KO}$ n = 15, $Nras^{G12V}/Pten^{KO}$ n = 9) determined from weekly/bi-weekly magnetic resonance imaging (MRI), and represented relative to each animal initial tumor volume.

**d** Representative images of H&E, Masson Trichrome, and Oil Red staining performed on sectioned livers collected from end-stage HCC-bearing mice. (Scale bars = 100 μm; representative of n = 4 mice). **e** Principal Component Analysis (PCA) plot depicting the transcriptome differences between genetically-distinct HCC bulk tumors ($Myc^{OE}/Trp53^{KO}$ n = 3, $Myc^{OE}/Pten^{KO}$ n = 3, $Nras^{G12D}/Pten^{KO}$ n = 4, $Nras^{G12V}/Pten^{KO}$ n = 3) and control livers injected with empty vectors, hereafter referred to as control (n = 3), following RNA-seq analyses (Supplementary Data 1). **f** Heatmap of unsupervised hierarchical clustering depicting the geneset enrichment of the MAPK, PI3K, and MYC signaling pathways in genetically-distinct HCCs ($Myc^{OE}/Trp53^{KO}$ n = 3, $Myc^{OE}/Pten^{KO}$ n = 3, $Nras^{G12D}/Pten^{KO}$ n = 4, $Nras^{G12V}/Pten^{KO}$ n = 3) compared to control (n = 3). The color scale represents the significance of the enrichment in $-\log_{10}$(FDR). FDR False Discovery Rate. Statistical significance was determined by log-rank (mantel-cox) test (**b**), two-sided hypergeometric test with multiple testing correction using Benjamini-Hochberg (**f**). Source data are provided as a Source Data file.

poorly-differentiated HCCs, while $Nras^{G12D}/Pten^{KO}$ and $Nras^{G12V}/Pten^{KO}$ tumors were moderately- and well-differentiated HCCs, respectively (Supplementary Fig. 1j), altogether confirming the correlation between tumor grade and overall survival reported in the human setting[4,61,62].

We next exposed the transcriptome profiles of genetically-distinct end-stage HCCs by performing RNA-seq on bulk tumors. Interestingly, both $Myc^{OE}$-driven models shared similar transcriptional profiles, supporting the dominant influence of MYC as the main oncogenic driver in these models, whereas $Nras^{G12V}/Pten^{KO}$ transcriptional signature resembled empty vector-injected control livers, likely due to their more differentiated features (Fig. 1e and Supplementary Fig. 1j, k). In contrast, $Nras^{G12D}/Pten^{KO}$ tumors clustered away from the other samples (Fig. 1e), indicative of its unique transcriptome profile. We further validated the states of the signaling pathways governed by the enforced oncogenic drivers (Fig. 1f, Supplementary Fig. 1l and Supplementary Data 1). Expectedly, $Myc^{OE}$ models exhibited increased activity in the MYC pathway, potentially overriding other more subtle changes related to $Pten$ loss-induced PI3K signaling activation in $Myc^{OE}/Pten^{KO}$ tumors. MAPK activity was heightened in both $Nras$-over-expressing models, albeit to a stronger extent in $Nras^{G12D}$-driven HCC (Fig. 1f and Supplementary Fig. 1a, l), suggesting differential MAPK intensity downstream of distinct $Nras$ point mutations. Altogether,

these results highlight the distinctive contribution of oncogenic pathways in influencing HCC progression and features, thereby providing relevant tools to study the impact of cancer cell-intrinsic signaling states on HCC multilayered complexity.

## Genetically-distinct murine HCC models correlate with distinct human HCC molecular subclasses and prognostic rates

The heterogeneity of human HCCs has recently been revisited to incorporate molecular subclasses[4]. In order to validate the relevance of our HCC models to the human pathology, we used the transcriptomic profiles of genetically-distinct tumors (Fig. 1e) to establish gene signatures specific to each HCC model (Supplementary Fig. 2a and Supplementary Data 1). We next applied a class prediction algorithm[26,63,64] to the normalized transcriptomes and assessed their resemblance to human HCCs[64–68]. Each of the genetically-distinct tumors segregated into different human molecular subtypes that matched their survival and histological features (Fig. 2a and Supplementary Fig. 2b). While $Myc^{OE}/Trp53^{KO}$, $Myc^{OE}/Pten^{KO}$, and $Nras^{G12D}/Pten^{KO}$ transcriptional profiles all correlated with the human HCC proliferation class, $Myc^{OE}$-driven HCCs closely associated with the human iCluster 3 and S2 classification and $Nras^{G12D}$-driven HCC fitted within the iCluster 1 and S1 subclasses. In sharp contrast, $Nras^{G12V}/Pten^{KO}$-driven tumors associated

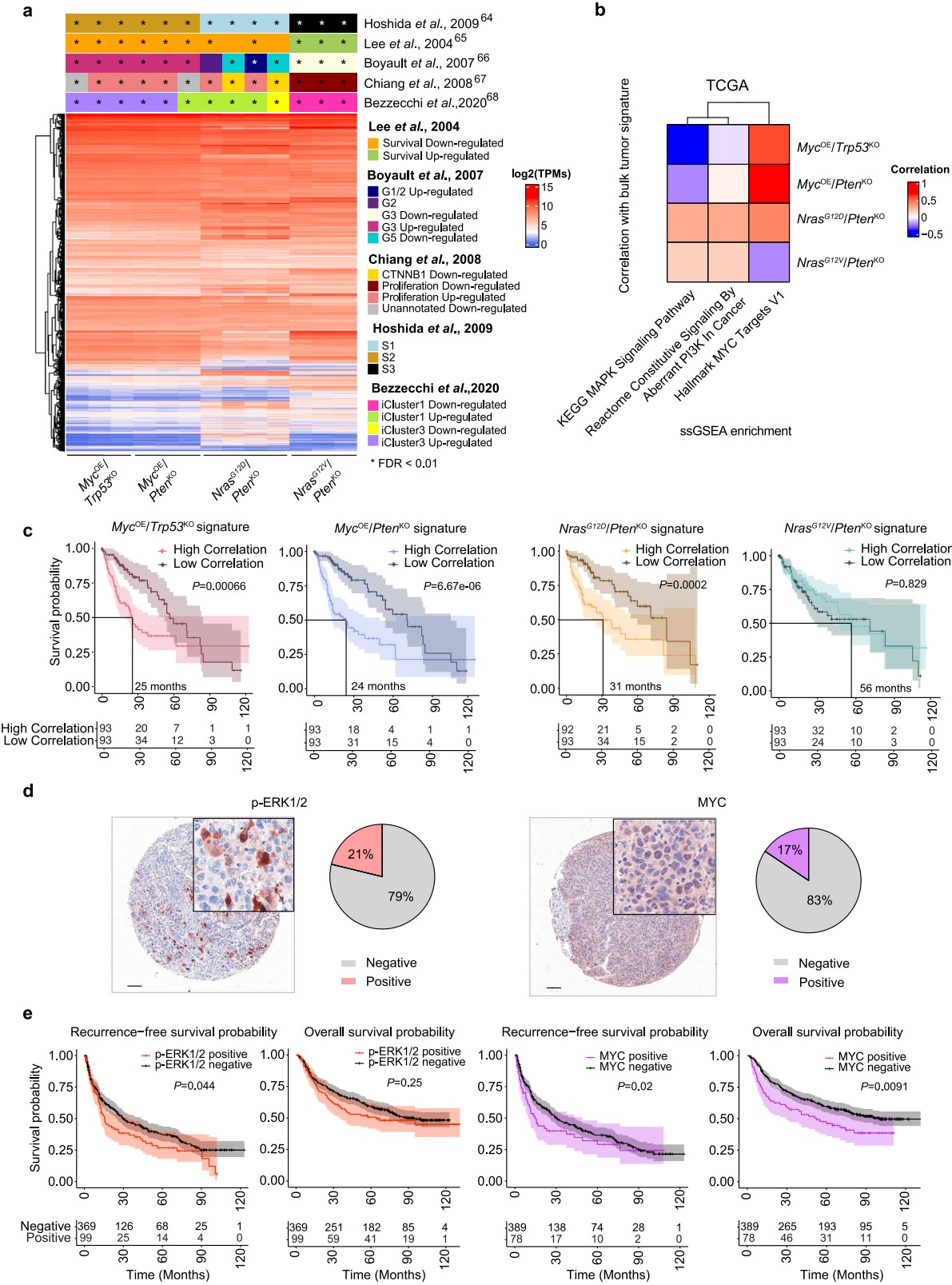

with the non-proliferation class of HCCs, which correlated with better patient survival, low proliferation signatures and tallied to the human S3 subclass of patients bearing hepatocyte-like, more indolent liver tumors[64] (Fig. 2a and Supplementary Fig. 2b). Altogether, these findings closely recapitulated the survival rates and histological features observed in vivo (Fig. 1b–d), with fast-growing, less differentiated $Myc^{OE}$- and $Nras^{G12D}$-driven models, while $Nras^{G12V}$-driven tumors

correspond to the well-differentiated human molecular classification exhibiting better prognosis (Fig. 2a). These findings underscore the effect of *Nras* point mutations in driving divergent HCC models, further singularizing $Nras^{G12D}$-driven HCC as a distinct aggressive model compared to $Myc^{OE}$-associated liver tumors.

We next sought to examine the enrichment profile of these oncogenic signaling pathways (Fig. 1f) across TCGA HCC patients and

**Fig. 2 | Transcriptional gene signatures derived from genetically-distinct HCC models recapitulate human HCC heterogeneity and predict patient prognosis.** **a** Heatmap of unsupervised hierarchical clustering depicting the global gene expression level as TPMs of genetically-distinct HCCs ($Myc^{OE}/Trp53^{KO}$ $n = 3$, $Myc^{OE}/Pten^{KO}$ $n = 3$, $Nras^{G12D}/Pten^{KO}$ $n = 4$, $Nras^{G12V}/Pten^{KO}$ $n = 3$). Top annotations represent the classification of HCC mouse models' transcriptomic profiles based on the human molecular HCC subtypes[64–68]. *FDR < 0.01. **b** Heatmap of unsupervised hierarchical clustering depicting the link between the MAPK, PI3K, and MYC signaling pathway enrichment (Supplementary Fig. 2c) and patients correlating to genetically-distinct HCC-derived transcriptional signatures across TCGA: Liver Hepatocellular Cancer (LIHC) patients ($n = 423$)[103] (Supplementary Data 2). **c** Kaplan–Meier survival curves displaying TCGA:LIHC patients[103] segregated according to their high/low correlation with the transcriptional signatures of each genetically-distinct HCC relative to control. Lines at survival probability = 0.5 depict median survival. Risk tables show the number of patients at the indicated time points (in months). **d** Representative IHC image for p-ERK1/2 and MYC performed on human HCC TMA sections from the Wu et al. dataset[69]. Pie charts depict the percentage of patients positive for p-ERK1/2 and MYC. (Scale bars = 100 μm; representative of $n = 99$ p-ERK1/2 positive patients and $n = 78$ MYC positive patients). **e** Kaplan–Meier curves displaying the overall survival (OS) and recurrence-free survival (RFS) of HCC patients (from the Wu et al. dataset[69]) segregated according to p-ERK1/2 negative ($n = 368$) or positive ($n = 99$) and c-MYC negative ($n = 389$) or positive ($n = 78$) staining in cancer cells. Risk tables show the number of patients at the indicated time points (in months) (see Supplementary Data 4 for median OS and RFS time). Statistical significance was determined by one-sided Fisher's test using Bonferroni multiple testing correction (**a**), log-rank test (**c**, **e**). The shading represent 95% confidence interval (**c**, **e**). TPMs Transcripts per Million. Source data are provided as a Source Data file.

interrogated their correlation with murine HCC-derived signatures (Fig. 2b and Supplementary Fig. 2c, d). These analyses asserted the dominance of *Myc*-driven transcriptional education in patients correlating with the $Myc^{OE}$-derived murine signatures, and overall heightened level of the MAPK signaling pathway in patient correlating with $Nras^{G12D}/Pten^{KO}$ compared to the $Nras^{G12V}/Pten^{KO}$ counterpart, in line with the results obtained in murine models. Moreover, the aggressive HCC models and human patients they associated with displayed matching overall survival outcome (Fig. 2c and Supplementary Data 2; see Methods for patient classification). The translational relevance of the murine HCC models was further supported by tumor mutation burden (TMB) (Supplementary Fig. 2e) and mutational pattern (Supplementary Fig. 2f and Supplementary Data 3) analyses from whole exome sequencing performed in each HCC murine model, which were overall comparable to Liver Hepatocellular Cancer (LIHC) patients.

To validate our findings in an independent cohort of HCC patients, we used a large tumor microarray dataset of 488 human HCCs[69]. These samples comprise mostly poorly to moderately-differentiated HCCs with hepatitis B infection being the most common etiology (Supplementary Data 4), thereby representing human tumors with similar clinical-histopathological features to $Nras^{G12D}/Pten^{KO}$- and $Myc^{OE}$-induced murine HCC. Analyses of oncogenic signaling activity using immunohistochemistry (IHC) staining revealed that MYC and MAPK/ERK were active in 17% and 21% of patients, respectively (Fig. 2d) and both correlated with poor patient prognosis (Fig. 2e). Altogether, our findings indicate that the activation state of oncogenic signaling pathway downstream of cancer cell mutations share similar prognostic rates in murine and human HCCs, both at the protein and transcriptome levels.

### Distinct oncogenic signaling pathways modulate the systemic and local tumor immune landscape

We next interrogated the impact of cancer cell genetic heterogeneity and related pathway activation on shaping the tumor immune landscape at the local and systemic levels during disease progression. Clear differences in the proportion of lymphoid and myeloid cell content were identified, with a significant dominance of myeloid cells in the more aggressive $Myc^{OE}/Pten^{KO}$, $Myc^{OE}/Trp53^{KO}$, and $Nras^{G12D}/Pten^{KO}$ HCCs (Fig. 3a). Longitudinal blood analyses of tumor-bearing mice suggested that this local increase partly originated from a systemic, tumor-induced myeloid cell response (Fig. 3b), with different myeloid cell subsets being mobilized in a cancer cell genetics-dependent manner (Supplementary Fig. 3a). While systemic neutrophilia was observed in all four models to varying extents, the proportion of monocyte subsets increase—classical Ly6C$^{high}$ versus the non-classical Ly6C$^{low}$ monocytes—was specific to the myeloid-enriched HCC models (Supplementary Fig. 3a).

Profiling of the TME at different stages of HCC progression in genetically-distinct models confirmed that immune cell composition and activation of lymphocytes and myeloid cells were tailored to the cancer cell mutational status at the bulk tumor level (Fig. 3c and Supplementary Fig. 3b). Irrespective of disease stage, T cells were scarce in the more aggressive, myeloid-enriched $Myc^{OE}$- and $Nras^{G12D}$-driven tumors. Contrastingly, $Nras^{G12V}/Pten^{KO}$-driven TME displayed an overall increased content and activation of effector T cells, emphasizing the striking differences enforced by distinct $Nras$ point mutations on the HCC TME. T cells were distinctively distributed in the HCC TME, with $Myc^{OE}$-driven tumors poorly infiltrated by CD3$^+$ cells, consistent with the immune-desert properties attributed to $Myc$ proto-oncogene[21,56] (Supplementary Fig. 3c). $Nras^{G12V}/Pten^{KO}$ nodules displayed an "inflamed/ immunological hot" phenotype[70], while $Nras^{G12D}/Pten^{KO}$ tumors presented CD3$^+$ cells restricted to the peritumoral and tumor edge regions (Supplementary Fig. 3c), a pattern referred to as "excluded/immunological cold"[70], suggesting the establishment of a pro-tumorigenic, immunosuppressive environment. Increased infiltration of neutrophils and Ly6C$^{high}$ monocytes was observed in $Myc^{OE}$ and $Nras^{G12D}$-driven tumors. These cells displayed low antigen presentation capacity and heightened immunosuppressive potential in all three aggressive models, albeit to a greater extent in $Nras^{G12D}$-driven tumors, as evidenced by MHCII and PD-L1 expression, respectively (Supplementary Fig. 3b). Moreover, $Myc^{OE}$- and $Nras^{G12D}$-driven tumors exhibited an increase in monocyte-derived macrophages (MDMs), with the latter HCC model showing the highest contribution of this population within the HCC TME (Fig. 3c and Supplementary Fig. 3d). This was in sharp contrast with $Nras^{G12V}$-driven tumors, where tissue-resident Kupffer cells (KCs) still composed the largest proportion of hepatic macrophages (Supplementary Fig. 3d). Furthermore, $Nras^{G12D}$-driven tumors exhibited the highest expansion of PD-L1$^+$ dendritic cells (DC) (Fig. 3c and Supplementary Fig. 3b), which likely contributes to a tumor-permissive microenvironment. Collectively, these results reveal the accumulation of infiltrating myeloid cells with heightened immunosuppressive capacity within the $Nras^{G12D}$-driven tumor milieu, thus setting this model apart from its $Nras^{G12V}$-driven counterpart, but also from $Myc^{OE}$-driven HCCs.

Crucially, comparable TME landscape alterations were associated to specific oncogenic signaling pathway activation identified in human HCC (Fig. 2d). Indeed, activation of both MAPK/ERK and MYC pathways correlated with an increased CD11B$^+$ myeloid cell presence and CD15$^+$ neutrophil recruitment (Fig. 3d), corroborating with our findings in $Myc^{OE}$- and $Nras^{G12D}$-driven murine models (Fig. 3a, b). Patients with heightened MAPK/ERK activity exhibited increased numbers of CD204$^+$ positive cells, which identifies macrophages with pro-tumorigenic functions[71,72]. In contrast, MYC-overexpressing patients showed increased infiltration of S100A9$^+$ immunosuppressive myeloid cells[73,74]. Thus, these results provide further evidence of the distinct myeloid cell education and poor patient prognosis (Fig. 2e) associated with the activation of Myc and MAPK/ERK pathways.

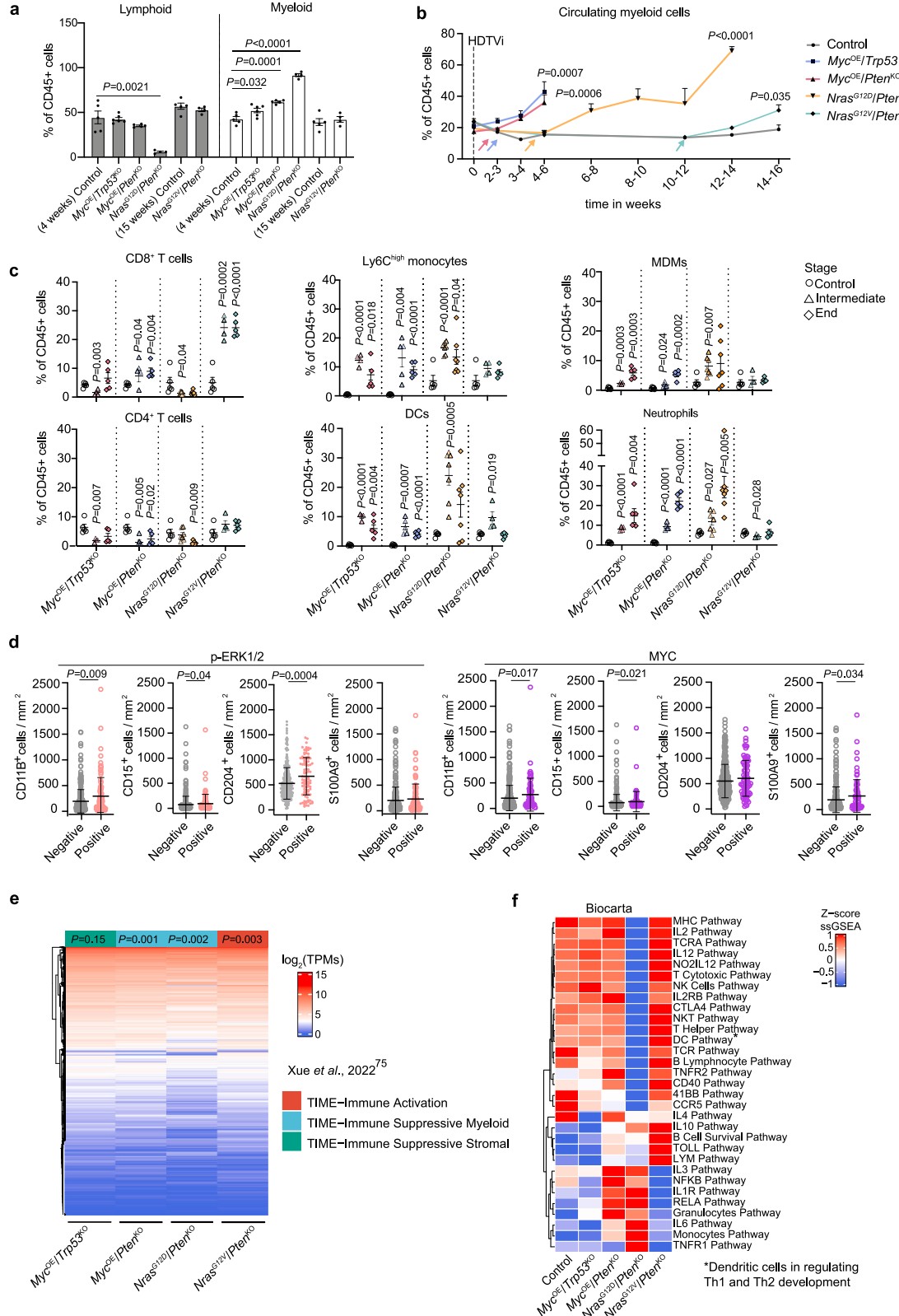

## Transcriptome profiling reveals a proinflammatory and tumor-permissive immune cell contexture within the $Nras^{G12D}/Pten^{KO}$ HCC TME

We next delved into the transcriptional heterogeneity of immune cells populating genetically-distinct HCC by performing RNA sequencing on FACS-isolated CD45⁺ cells (Supplementary Fig. 3e, f and Supplementary Data 5). To unbiasedly characterized the TME phenotype

across the distinct models, we classified their transcriptome profiles according to recently published liver tumor immune microenvironment (TIME) subtypes[75]. The $Nras^{G12V}/Pten^{KO}$ TME significantly classified in the TIME-IA (Immune Active) subtype (Fig. 3e), associated with significant T cell activation and infiltration, in line with our previous findings (Fig. 3c and Supplementary Fig. 3b). Distinctively, $Myc^{OE}/Pten^{KO}$ and $Nras^{G12D}/Pten^{KO}$ TME tallied to the TIME-SM (Suppressive

**Fig. 3 | Distinct oncogenic drivers' activation impact HCC immune landscape.**
**a** Myeloid and lymphoid cell contents relative to total CD45$^+$ leukocytes in end-stage HCC and age-matched control livers Control-4-weeks $n = 5$, $Myc^{OE}/Trp53^{KO}$ $n = 6$, $Myc^{OE}/Pten^{KO}$ $n = 5$, $Nras^{G12D}/Pten^{KO}$ $n = 4$, Control-15-weeks $n = 5$, and $Nras^{G12V}/$ $Pten^{KO}$ $n = 4$). **b** Quantification of circulating myeloid cells relative to total CD45$^+$ leukocytes at the indicated timepoints for each of the HCC models (Control $n = 10$, $Myc^{OE}/Trp53^{KO}$ $n = 20$, $Myc^{OE}/Pten^{KO}$ $n = 18$, $Nras^{G12D}/Pten^{KO}$ $n = 30$, $Nras^{G12V}/Pten^{KO}$ $n = 16$). Arrows indicate timepoints when tumors were detectable. Statistical significance: final analyzed blood samples of tumor-bearing mice versus aged-matched controls. **c** Percentage of the immune cell populations relative to total CD45$^+$ leukocytes in control and HCCs at intermediate and end-stage (Control-4-weeks $n = 5$, $Myc^{OE}/Trp53^{KO}$ intermediate $n = 4$ and end-stage $n = 5$, $Myc^{OE}/Pten^{KO}$ intermediate $n = 5$ and end-stage $n = 5$, Control-15-weeks $n = 5$, $Nras^{G12D}/Pten^{KO}$ intermediate $n = 7$ and end-stage $n = 7$ (for CD8T and CD4T $n = 6$); and $Nras^{G12V}/$ $Pten^{KO}$ intermediate $n = 4$ and end-stage $n = 5$. Statistical analyses of each model are performed comparing each tumor stage to its relative control group (Control 4-weeks for $Myc^{OE}/Trp53^{KO}$ and $Myc^{OE}/Pten^{KO}$; and Control-15-weeks for $Nras^{G12D}/$ $Pten^{KO}$ and $Nras^{G12V}/Pten^{KO}$). **d** Quantification of CD11B, CD15, CD204 and S100A9

positive cells by immunohistochemistry in paraffin-embedded HCC patient samples from the Wu et al. dataset[69] segregated according to p-ERK1/2 (positive $n = 99$, negative $n = 369$ for CD11B, CD15 and CD204; positive $n = 98$, negative $n = 367$ for S100A9) and MYC (positive $n = 78$, negative $n = 389$ for CD11B, CD15 and CD204; positive $n = 77$, negative $n = 386$ for S100A9) expression in cancer cells. **e** Unsupervised hierarchical clustering of the transcriptome of CD45$^+$ cells isolated from genetically-distinct HCCs ($n = 3$ for all genotypes) and their classification according to the human HCC immune subtypes clustering[75]. **f** Unsupervised hierarchical clustering of the ssGSEA enrichment scores per control liver and HCC models (end-stage) using immune-related pathways presented in the Biocarta database. The color scale represents the z-score normalized enrichment per pathway (row) between HCC models (Control $n = 5$, $Myc^{OE}/Trp53^{KO}$ $n = 3$, $Myc^{OE}/$ $Pten^{KO}$ $n = 3$, $Nras^{G12D}/Pten^{KO}$ $n = 4$, $Nras^{G12V}/Pten^{KO}$ $n = 5$). Graphs show mean ± SEM (**a**, **c**, **d**), +SEM (**b**). Statistical significance was determined by unpaired two-sided Student's $t$-test (**a**–**c**), two-sided Mann-Whitney U test (**d**), one-sided Fisher's test using Bonferroni multiple testing correction (**e**). See Supplementary Fig. 9j, k for gating strategy (**a**–**c**). Source data are provided as a Source Data file.

Myeloid) subtype (Fig. 3e), associated with poor patient prognosis, myeloid cell dominance, and increased expression of both immuno-suppressive and IL1-related inflammatory signaling pathways. Moreover, human HCC samples displaying high MAPK/ERK or MYC pathway activation were over-represented in patients exhibiting a stronger enrichment of the prognostic myeloid signature Myeloid Response Score (MRS, Supplementary Fig. 3g), which correlates with immuno-suppressive and pro-tumorigenic TME features[69]. Altogether our findings indicate that MYC overexpression and MAPK/ERK pathway activation in cancer cells drives a tumor-permissive, myeloid-enriched TME in both preclinical and human HCC.

Next, the education profiles of CD45$^+$ cells were further analyzed using immune clusters previously described in ref. 76, as well as by comparing their transcriptome profiles to additional immune-related pathways (Fig. 3f and Supplementary Data 6). Gene sets enriched in $Myc^{OE}$-driven models encompassed increased proliferation signature, Th2 cells and Interferon Gene clusters (Fig. 3f and Supplementary Fig. 3h). Protein-protein interaction enrichment analyses identified the latter to be related to type I interferon signaling (Supplementary Fig. 3i), often associated with innate immune cell response[77]. Conversely, lymphoid-rich, slow-growing $Nras^{G12V}$-driven tumors exhibited a significant enrichment for T cell gene sets, Angiogenesis and the Immunologic Constant of Rejection score (Supplementary Fig. 3h), generally related to Th1 immunity[78], as well as increased IL-2, T cell activation, and co-stimulatory pathways (CD40, 41BB) (Fig. 3f), alto-gether confirming heightened adaptive immune response in these tumors. Substantiating the flow cytometry analyses and further sin-gling out the unique TME profile of these tumors (Supplementary Fig. 3e, f), $Nras^{G12D}/Pten^{KO}$ HCC displayed an enrichment of macro-phages, immature DCs, TGF-β signaling and Neutrophils pathway activity (Supplementary Fig. 3h). In line with the immunosuppressive profile observed within the myeloid compartment (Fig. 3e and Sup-plementary Fig. 3b), T cell-related signaling pathways associated with activation and function were down-regulated in $Nras^{G12D}/Pten^{KO}$ immune cells (Fig. 3f). Concomitantly, $Nras^{G12D}/Pten^{KO}$ displayed an enrichment in several proinflammatory pathways with well-described HCC pro-tumorigenic features[6] also characteristic of the TIME-SM human subtype, such as NF-κB, IL1R, IL6, and TNFR1 (Fig. 3f). As these pathways are known to be related to inflammasome activity[79,80], we evaluated the levels of active caspase-1 in bulk tumors, which was significantly increased in $Nras^{G12D}/Pten^{KO}$ HCC (Supplementary Fig. 3j).

Altogether, these observations highlight the distinct HCC immune landscapes associated with different cancer cell molecular profiles. Despite a shared enrichment in myeloid cell content amongst the more aggressive tumor models, we consistently identified diverse education profiles between $Myc^{OE}$-driven and $Nras^{G12D}/Pten^{KO}$ HCCs. Indeed, the

latter exhibited a unique inflammatory background with prominent immunosuppressive capacity and dampened T cell activation features.

### $Nras^{G12D}/Pten^{KO}$-associated myeloid cells display a prominent pro-tumorigenic, inflammatory signature at the single cell level

We further explored the heterotypic interactions underlying the unique $Nras^{G12D}$-driven HCC TME by performing single-cell RNA sequencing (scRNA-seq) on end-stage $Nras^{G12D}/Pten^{KO}$-driven tumor and age-matched control livers. Differential expression (DE) analysis identified seven unique clusters defined as immune cells based on $Ptprc$ (CD45) expression (Fig. 4a and Supplementary Fig. 4a–c) and then classified as myeloid or lymphoid subsets according to their transcriptional profiles (Supplementary Fig. 4b–d and Supplementary Data 7). We determined that both myeloid cell subsets - monocytic cells and neutrophils—were significantly enriched in $Nras^{G12D}/Pten^{KO}$ HCC, while lymphoid cells (T cell, NK cells and B cells), pDCs and cDC1s were decreased compared to control (Fig. 4b and Supplementary Fig. 4c), emphasizing the major contribution of the myeloid com-partment to the $Nras^{G12D}/Pten^{KO}$ TME at the single-cell resolution.

In light of the prominent inflammatory features observed in $Nras^{G12D}/Pten^{KO}$ CD45$^+$ cells and bulk tumors (Fig. 3f and Supplementary Fig. 3j), we next investigated the different immune-modulating func-tions altered within $Nras^{G12D}/Pten^{KO}$ TME cell subsets. Gene set analysis of $Nras^{G12D}/Pten^{KO}$ up-regulated genes showed significant enrichment for 'NF-κB signaling' and 'Inflammatory response' in the 'Monocytic cell' cluster (Supplementary Fig. 4e and Supplementary Data 8). Upon assessment of the genes included in these pathways, we observed a significant increase in proinflammatory ($Il1a$, $Il1b$ and $Nlrp3$[79,81,82]) and immunosuppressive ($Nos2$, $Arg1$ and $Vegfa$[83–85]) gene expres-sion (Fig. 4c).

We next explored the 'Monocytic cells' subset education profile, which we hypothesized likely contributed to the dual inflammatory and immunosuppressive phenotype of $Nras^{G12D}/Pten^{KO}$ HCC TME. Fol-lowing the subsampling of this cluster, we identified nine subpopula-tions (Fig. 4d), from which the 'C1_Monocytic_$March3^+$', 'C2_Classical monocytes_$Chil3^+$', 'C3_Monocytic_$C1qa^+$' and 'C6_Monocytic_$Cxcl3^+$' were significantly more abundant in $Nras^{G12D}/Pten^{KO}$ compared to control (Fig. 4e and Supplementary Fig. 4f). Unbiased cluster annota-tion using a publicly available human single-cell RNAseq-dataset[75] revealed that most monocytic subclusters exhibited macrophage, monocytes and DCs signatures (Supplementary Fig. 4g). DE analysis between control and $Nras^{G12D}/Pten^{KO}$ clusters within the 'Monocytic cell' population identified $Il1b$ and $Nlrp3$ up-regulation across all sub-populations (Fig. 4f), confirming the involvement of the inflamma-some and IL-1 pathways in $Nras^{G12D}/Pten^{KO}$ tumorigenesis. Moreover, expression of immunosuppressive genes, such as $Arg1$ and $Vegfa$, was

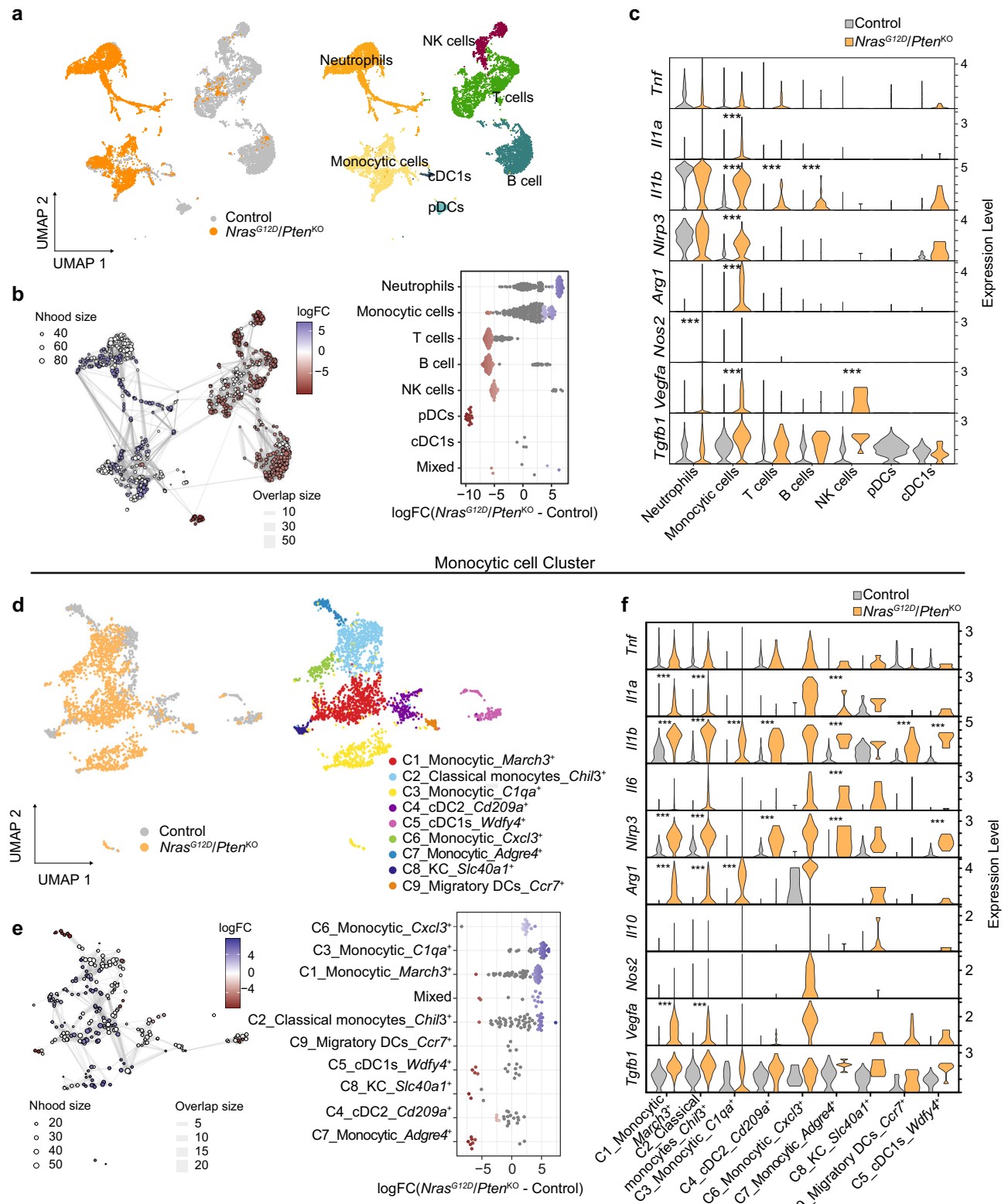

increased in the 'C1_Monocytic_*March3*+', 'C2_Classical monocytes_*Chil3*+' and 'C3_Monocytic_*C1qa*+ clusters (Fig. 4f). These findings were independently validated by RT-qPCR analyses comparing FACS-isolated MDMs and Ly6C^high monocytes from control livers and *Nras*^G12D/*Pten*^KO HCC (Supplementary Fig. 4h), which likely comprise the majority of cells identified within the 'Monocytic cells' subcluster. The increase in *Il1a*, *Il1b* and *Nlrp3* mRNA levels at different stages of *Nras*^G12D/*Pten*^KO progression indicate that the IL-1 and

inflammasome pathways are important attributes of myeloid cell education. Additionally, *Arg1* was also significantly increased in these myeloid cell populations (Supplementary Fig. 4h), evoking their immunosuppressive phenotype, as previously reported in *Kras*^G12D-driven pancreatic adenocarcinoma (PDAC) tumors[86]. Collectively, these results highlight the heterogeneity of the *Nras*^G12D-dictated TME, exposing a myeloid-rich immune landscape displaying mixed inflammatory and pro-tumorigenic features.

**Fig. 4 | The *Nras*^G12D/*Pten*^KO TME displays transcriptomic heterogeneity with mixed pro-tumorigenic and inflammatory signatures. a** Uniform Manifold Approximation and Projection (UMAP) representation of CD45$^+$ immune cells from control liver ($n = 1$ mouse; 7447 cells) and *Nras*^G12D/*Pten*^KO HCC ($n = 1$ mouse; 4814 cells) (left) with annotated populations identified by scRNA-seq (right, Supplementary Data 7). **b** K-nearest neighbor (KNN) graph (left) and dotplot (right) depicting the differential abundance of cell types between *Nras*^G12D/*Pten*^KO HCC relative to control. Each dot represents a group of cells clustered in 'neighborhoods'. Colors represent significant logFC (FDR ≤ 0.05), whereas white is non-significant difference in abundance. Edge thickness represents the number of overlapping cells between neighborhoods. **c** Violin plots depicting the normalized expression levels of inflammatory and immunosuppressive genes in the indicated immune cell subsets in control liver ($n = 1$) and *Nras*^G12D/*Pten*^KO HCC ($n = 1$)

(Supplementary Data 8). **d** UMAP representation of the 'Monocytic cell' subset (2616 cells) grouped with cDC1 (115 cells) from (a). **e** K-nearest neighbor (KNN) graph (left) and dotplot (right) depicting the differential abundance of myeloid cell subsets between *Nras*^G12D/*Pten*^KO HCC and control liver from cells grouped as 'neighborhoods' in (d). The colored dots represent significant changes in abundance using a threshold of FDR ≤ 0.05, whereas white is non-significant difference in abundance. Edge thickness represents the number of overlapping cells between neighborhoods. **f** Violin plots depicting the normalized expression levels of inflammatory and immunosuppressive genes in the indicated myeloid cell subsets in control livers and *Nras*^G12D/*Pten*^KO HCC from the 'Monocytic cell' population (Supplementary Data 8). Nhood = neighborhood. Statistical significance was determined by two-sided, Wilcoxon rank sum test with Bonferroni multiple testing correction (**c**, **f**).*** FDR ≤ 0.001. Source data are provided as a Source Data file.

## GM-CSF signaling is uniquely activated in *Nras*^G12D-driven tumors

We next sought to investigate the cancer cell-derived molecular players underlying *Nras*^G12D/*Pten*^KO unique TME profile, and carried out RNA-seq of *Nras*^G12D/*Pten*^KO and *Nras*^G12V/*Pten*^KO HCC cell lines to complement bulk tumor analyses (Fig. 5a and Supplementary Data 9). Expectedly, gene expression profiles of *Nras*^G12D and *Nras*^G12V cancer cells clustered away from each other (Supplementary Fig. 5a), with changes largely conserved at the bulk tumor level (Fig. 5a). Gene set enrichment analyses of common DEG revealed that distinct lipid-associated pathways were significantly higher in *Nras*^G12V cancer cells (Supplementary Fig. 5b), consistent with the steatotic features observed in *Nras*^G12V/*Pten*^KO HCC (Fig. 1d). Processes related to inflammation, such as IL-2/STAT5, TNF-alpha signaling via NF-κB and inflammatory response were significantly higher in *Nras*^G12D/*Pten*^KO cancer cells and bulk tumors compared to their *Nras*^G12V-driven counterparts (Fig. 5b). Altogether, these results suggest that cancer cell-intrinsic features shape several of the *Nras*-driven tumor characteristics, including the myeloid cell proinflammatory education identified in the CD45$^+$ immune cells (Fig. 3f) and single-cell RNA-seq analyses (Fig. 4) of *Nras*^G12D/*Pten*^KO HCC.

We next interrogated the distinctive TME features of all four HCC models at the protein level by performing cytokine profiling of end-stage bulk tumors and cancer cell lines (Supplementary Fig. 5c, d). The secretome profiles of the two *Nras*-driven HCC models were then cross-checked with the up-regulated target genes identified by RNA sequencing. These unbiased analyses identified nine candidate factors up-regulated in *Nras*^G12D/*Pten*^KO bulk tumors and cancer cells both at the transcriptome and proteome levels (Fig. 5c). Among these, *Csf2*/GM-CSF was of particular interest, with previous studies highlighting the increased levels of this cytokine in *Kras*^G12D-driven PDAC associated with immature myeloid cell infiltration[86]. Indeed, GM-CSF is a master regulator of innate immune cells mediating their recruitment, survival, and activation[87]. Crucially, GM-CSF can promote a proinflammatory response, for instance by directly inducing the expression and secretion of IL-1 cytokines[87–89].

We thus hypothesized that the proinflammatory, myeloid-rich profile observed in the *Nras*^G12D/*Pten*^KO TME may be driven by a cancer cell-intrinsic regulation of GM-CSF. Increased secretion of GM-CSF was independently confirmed in the supernatant of *Nras*^G12D/*Pten*^KO cancer cells (Fig. 5d and Supplementary Fig. 5e) and at the transcriptomic and protein levels (Supplementary Fig. 5f–i). Importantly, by applying a publicly available GM-CSF signature[89] (Supplementary Data 9) to *Nras*^G12D/*Pten*^KO bulk tumor and CD45$^+$ RNA-seq datasets (Supplementary Data 9), we confirmed the significant enrichment of GM-CSF downstream pathways in this HCC model (Fig. 5e). Several cytokines and chemokines (e.g., *Ccl17*, *Cxcl14*, *Ccl24*, and *Ccl6*) were up-regulated in CD45$^+$ immune cells from *Nras*^G12D/*Pten*^KO tumors compared to the *Nras*^G12V/*Pten*^KO TME (Fig. 5f and Supplementary Fig. 5j). Interestingly, GM-CSF increase was specific to the tumor bed, and no systemic upregulation was observed in the peripheral blood (Supplementary

Fig. 5k), suggesting a local re-education of the myeloid micro-environment rather than a systemic-elicited response.

We next queried the scRNA-seq dataset of *Nras*^G12D/*Pten*^KO HCC (Fig. 4) to identify cell populations involved in GM-CSF signaling pathway. First, we validated that *Csf2* expression is virtually absent in stromal cells within the *Nras*^G12D/*Pten*^KO TME (Supplementary Fig. 5l), indicating that cancer cells are the major source of GM-CSF within the TME. Interestingly, while *Csf2ra*, *Csfr2b* and *Ccl6* were broadly expressed in myeloid cells (Fig. 5g), the GM-CSF gene expression signature was highly enriched in the two dominants 'Monocytic cell' and 'Neutrophils' myeloid cell clusters (Fig. 5h and Supplementary Fig. 5m, n). These results suggest a pivotal role of cancer cell-derived GM-CSF signaling in shaping the immune landscape of *Nras*^G12D-driven HCC.

Lastly, we investigated whether the *Nras*^G12D/*Pten*^KO-specific transcriptome and the GM-CSF signature may be of relevance in human HCC malignant features. First, we compared TCGA:LIHC patients presenting high correlation with either the *Nras*^G12D/*Pten*^KO or *Nras*^G12V/*Pten*^KO gene expression signatures (Fig. 2c) and determined that the former predicted poorer HCC patient survival (Fig. 5i). Patients with high *Nras*^G12D/*Pten*^KO correlation scored significantly higher than *Nras*^G12V/*Pten*^KO-associated patients for GM-CSF signature enrichment (Fig. 5j and Supplementary Fig. 5o). Furthermore, patients displaying high correlation with the GM-CSF signature exhibited a poorer prognosis (Fig. 5k and Supplementary Fig. 5p), implying that *Nras*^G12D/*Pten*^KO and GM-CSF transcriptional signatures are tightly connected in human HCCs and predict worse disease outcome. Overall, these results identified GM-CSF as a potential therapeutic target in subgroups of HCC patients, while raising the questions of the underlying molecular mechanisms regulating GM-CSF secretion specifically in *Nras*^G12D/*Pten*^KO.

## Increased ERK1/2 activity drives GM-CSF secretion in a SP1-dependent manner and correlates with GM-CSF signature at the pan-cancer level

As GM-CSF is uniquely secreted by *Nras*^G12D/*Pten*^KO cells (Fig. 5d), we sought to identify the molecular players downstream of Ras/MAPK and PI3K pathways involved in this specific regulation. We observed an increase in all phosphorylated MAPK-associated proteins in *Nras*^G12D/*Pten*^KO compared to *Nras*^G12V/*Pten*^KO cancer cells, including ERK1/2, AKT, S6RP, 4EBP1 and c-Jun (Fig. 6a and Supplementary Fig. 6a, b), further validating the differential intensity of MAPK signal activity between the two *Nras*-driven HCC models. In order to identify which MAPK-associated proteins governed GM-CSF secretion in *Nras*^G12D/*Pten*^KO cells, we used several inhibitors targeting RAS effector proteins, namely MEK (Trametinib), ERK1/2 (Temuterkib), ERK2 (Vx-11), mTOR (AZD8055), AKT (MK2066) and JNK (SP600125) (Supplementary Fig. 6c, d). While inhibition of mTOR, AKT and JNK pathways did not affect GM-CSF level (Supplementary Fig. 6d), MEK1/2, ERK1/2, and ERK2 inhibitors hampered *Nras*^G12D/*Pten*^KO-derived GM-CSF secretion (Fig. 6b) with no measurable effects on cancer cell proliferation

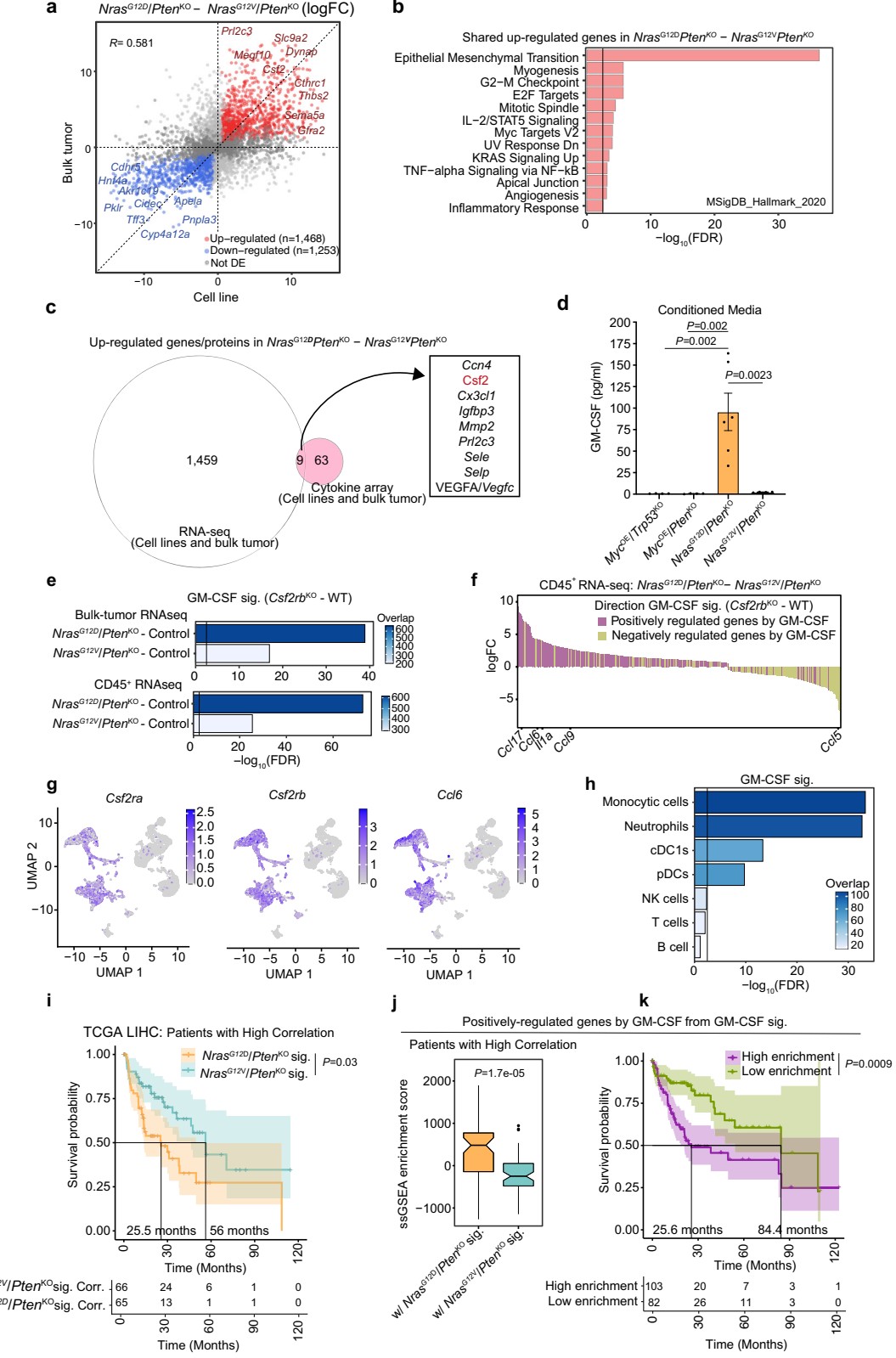

(Supplementary Fig. 6e). We further validated this finding by generating constitutive *Erk1* (sh*Erk1*) and *Erk2* (sh*Erk2*) knockdown cell lines (Supplementary Fig. 6f) and assessed the levels of GM-CSF in their supernatants (Fig. 6c). The significant decrease in GM-CSF secretion observed in these cells in absence of proliferation defects (Supplementary Fig. 6g) further confirmed the central role of the ERK1/2 signaling pathway activity in driving GM-CSF secretion.

We next sought to identify ERK1/2 downstream factors regulating *Csf2* expression by performing RNA-seq on both *Nras*[G12D]/*Pten*[KO] and *Nras*[G12V]/*Pten*[KO] cells upon ERK signaling blockade (Supplementary Fig. 6h and Supplementary Data 9). We queried this dataset for changes unique to *Nras*[G12D] point mutation following ERK inhibition (Fig. 6d) and selected promoters regulating genes which expression profiles followed the same trajectory than *Csf2* in the different samples

**Fig. 5 | Nras^G12D/Pten^KO cancer cells activate GM-CSF signaling in the TME.**
**a–c** Differential expression analyses from bulk tumor (Fig. 1e) and cancer cell RNA-seq datasets comparing Nras^G12D/Pten^KO (n = 3) and Nras^G12V/Pten^KO HCC (n = 3). **a** DEGs between bulk tumors (y-axis) and cancer cells (x-axis). Colored dots represent genes significantly deregulated in both datasets. **b** Enriched signaling pathways for genes in (**a**). **c** Overlap of up-regulated genes and proteins identified by RNA-sequencing and cytokine arrays, respectively. **d** GM-CSF levels in conditioned media of genetically-distinct HCC cells (Myc^OE/Trp53^KO n = 4, Myc^OE/Pten^KO n = 4, Nras^G12D/Pten^KO n = 6, Nras^G12V/Pten^KO n = 4). **e** GM-CSF signature[89] enrichment (Csf2rb^KO-WT) in DEG from transcriptional signatures of Nras^G12D/Pten^KO (n = 4) and Nras^G12V/Pten^KO (n = 3) bulk tumors (top), and Nras^G12D/Pten^KO (n = 3) and Nras^G12V/Pten^KO (n = 3) CD45^+ cells (bottom). Colors represent DEG overlap from Nras-driven models and GM-CSF signature. **f** DEGs from CD45^+ cell RNA-seq of Nras^G12D/Pten^KO (n = 7) relative to Nras^G12V/Pten^KO (n = 7) overlapping with GM-CSF signature[89]. Colors represent direction of GM-CSF regulation, relevant genes from GM-CSF signature are depicted (Supplementary Data 9). **g** Csf2ra, Csf2rb and Ccl6 expression from scRNA-seq (Fig. 4a). **h** GM-CSF signature[89] enrichment for the indicated immune cell subsets identified by scRNA-seq. **i** Survival of TCGA:LIHC patients[103] segregated into high correlation with Nras^G12D/Pten^KO or Nras^G12V/Pten^KO transcriptional signatures (Fig. 2c) (Nras^G12D/Pten^KO n = 65, Nras^G12V/Pten^KO n = 66). high- and low-correlating patients are unique to each signature. Tables show patient numbers at the indicated time points. **j** Enrichment of GM-CSF positively-regulated genes (n = 628) from the GM-CSF signature[89] in TCGA:LIHC patients[103] segregated into high correlation with Nras^G12D/Pten^KO or Nras^G12V/Pten^KO transcriptional signatures (Fig. 2c) (Nras^G12D/Pten^KO n = 65, Nras^G12V/Pten^KO n = 66). **k** Survival of TCGA:LIHC patients[103] segregated into high/low enrichment of GM-CSF positively-regulated genes (n = 628) from the GM-CSF signature[89] (High n = 103, Low n = 82). Tables show patient numbers at the indicated time points. Graph show mean ± SEM (**d**). Statistical significance: two-sided hypergeometric test with Benjamini-Hochberg multiple testing correction (**b, e, h**), one-way ANOVA with Tukey's multiple comparison test (**d**), log-rank test (**i, k**), unpaired two-sided Student's t-test (**j**). Vertical lines at −log₁₀(FDR) = 2.5 indicate significance threshold (**b, e, h**). The shading represents 95% confidence interval (**i, k**). Source data are provided as a Source Data file.

analyzed (Fig. 6e). Next, we applied motif enrichment analysis on these promoters and identified the motif associated with the transcription factor SP1 as a top candidate (Fig. 6f and Supplementary Data 9). To assess the role of SP1 in modulating Csf2 expression and GM-CSF secretion, we generated a Sp1 knockdown cell line (shSp1) from the parental Nras^G12D/Pten^KO cells (Supplementary Fig. 6i). Both GM-CSF gene expression and cytokine secretion were hindered in shSp1 Nras^G12D/Pten^KO cells (Fig. 6g and Supplementary Fig. 6j), positioning SP1 as an important regulator of the Nras^G12D/MAPK-ERK/GM-CSF activation cascade. Finally, we validated the translational relevance of ERK1/2 and GM-CSF pathway correlation by assessing their transcriptional signature enrichment across all TCGA patients. Remarkably, the positive association between ERK1/2 pathway activity and GM-CSF signature was evident across several cancer types (Fig. 6h). Overall, this finding places ERK1/2 signaling node at the center of Nras^G12D-driven GM-CSF secretion, while underscoring the broader impact of MAPK-ERK and GM-CSF axes at the pan-cancer level.

### Nras^G12D/Pten^KO-derived GM-CSF promotes the accumulation of proinflammatory monocyte-derived Ly6C^low myeloid cells

We next investigated the role of Nras^G12D/Pten^KO-derived GM-CSF in shaping myeloid cell differentiation and phenotype in vitro. We differentiated bone marrow (BM) cells using conditioned media (CM) prepared from Nras^G12D/Pten^KO or Nras^G12V/Pten^KO cancer cell lines (Supplementary Fig. 7a), while recombinant M-CSF and GM-CSF were used for comparison. Interestingly, Nras^G12D/Pten^KO CM specifically led to an accumulation of Ly6C^lowCD11b^highF4-80^low cells, referred to as Ly6C^low herein (Fig. 7a and Supplementary Fig. 7b) while other myeloid cell subset contents were largely unchanged (Supplementary Fig. 7c). Analysis of the polarization profile of Nras^G12D/Pten^KO-induced Ly6C^low cells showed an enrichment of the immunosuppressive markers CD39 and PD-L1, a feature comparable to recombinant GM-CSF exposure (Fig. 7b) and corroborating with the phenotype of myeloid cells present within Nras^G12D/Pten^KO HCC (Supplementary Fig. 3b). In addition, the antigen presenting cell (APC)-related proteins CD80 and MHCII were similarly up-regulated (Fig. 7b), as previously reported in the context of GM-CSF stimulation[90]. Most importantly, Nras^G12D/Pten^KO CM-induced Ly6C^low cell abundance was abrogated upon GM-CSF neutralization (a-GM-CSF) (Fig. 7a), suggesting a key role of GM-CSF in supporting the survival and maintenance of this cell population. Moreover, the percentage of CD39^+ and PDL1^+ cells exhibited a reduction of five- and two-fold, respectively, in Ly6C^low cell populations in the context of GM-CSF blockade (Fig. 7b). We next investigated whether Nras^G12D/Pten^KO CM led to gene expression changes in BM cells towards a more proinflammatory profile, as highlighted by the scRNA-seq changes within the Nras^G12D/Pten^KO TME (Fig. 4c, f). We first confirmed the increased expression of GM-CSF downstream genes

Ccl6 and Ccl17 in BM cells differentiated with Nras^G12D/Pten^KO CM, which was abrogated upon GM-CSF neutralization (Supplementary Fig. 7d). Similarly, expression of the proinflammatory genes Il1a and Il6 were hindered in the context of GM-CSF blockade (Supplementary Fig. 7d). Moreover, BM cells cultured in Nras^G12D/Pten^KO CM displayed high IL-1R signaling activity which was reverted upon GM-CSF neutralization (Fig. 7c). Overall, these findings validate Nras^G12D/Pten^KO cancer cell-derived GM-CSF as a central regulatory cytokine promoting the differentiation and re-education of Ly6C^low myeloid cells with mixed immunosuppressive and proinflammatory/APC-like features.

We next interrogated whether similar myeloid cell content and activation changes were recapitulated in vivo in the context of GM-CSF blockade (Fig. 7d). As observed in vitro, a-GM-CSF treatment led to a sustained decrease in the content of Ly6C^low myeloid cells within the Nras^G12D/Pten^KO TME, observed as early as 2 weeks post treatment (time point: T1) and maintained in end-stage tumor-bearing mice (time point: T2) (Fig. 7e and Supplementary 7e), whereas the content of lymphocytes was not altered (Supplementary Fig. 7f). No systemic changes were observed in the peripheral myeloid cell content (Supplementary Fig. 7g), suggesting that the effect of GM-CSF on reshaping the myeloid landscape is primarily local. Moreover, GM-CSF inhibition curbed the proliferation of Ly6C^low cells in the Nras^G12D/Pten^KO TME (Fig. 7f), indicating that GM-CSF promotes the local expansion of this population. Furthermore, GM-CSF blockade compromised the mixed immunosuppressive/APC-like phenotype of Ly6C^low cells, as indicated by the decrease of cells expressing the immunosuppressive markers CD39 and PD-L1, as well as the APC proteins CD80, MHCII, and CD11c, specifically in the early phase of treatment (T1) (Supplementary Fig. 7h). Altogether, these results suggest that GM-CSF blockade dampens the abundance and transiently alters the phenotype of Ly6C^low cells both in vitro and in vivo.

To investigate the cell of origin and further scrutinize the phenotype and function of the GM-CSF-driven Ly6C^low myeloid subset in Nras^G12D/Pten^KO HCC, we probed for markers associated with distinct myeloid-derived cell types. FlowSOM analyses indicate that Ly6C^low cells comprised two dominant subsets, both characterized by heightened expression of CD206 and CCR2: CD206^highCCR2^high and CD206^intCCR2^high cells (Fig. 7g, h). Expression of CCR2 infers that the Ly6C^low pool is derived from a classical monocyte parental population[91]. The discrepant expression of CD206, a hallmark of immature DCs and a pro-tumorigenic macrophage marker[92–95], together with distinct levels of the APC proteins MHCII and CD11c[96], further highlight the mixed identity of the Ly6C^low pool presenting DC- and macrophage-like features (Fig. 7g, h). These results were corroborated by analyses of the C1_Monocytic_March3^+ and C6_Monocytic_Cxcl3^+ subclusters that significantly expanded in the Nras^G12D/Pten^KO TME (Fig. 4d, e), with both populations expressing Ccr2 and Mrc1 (CD206),

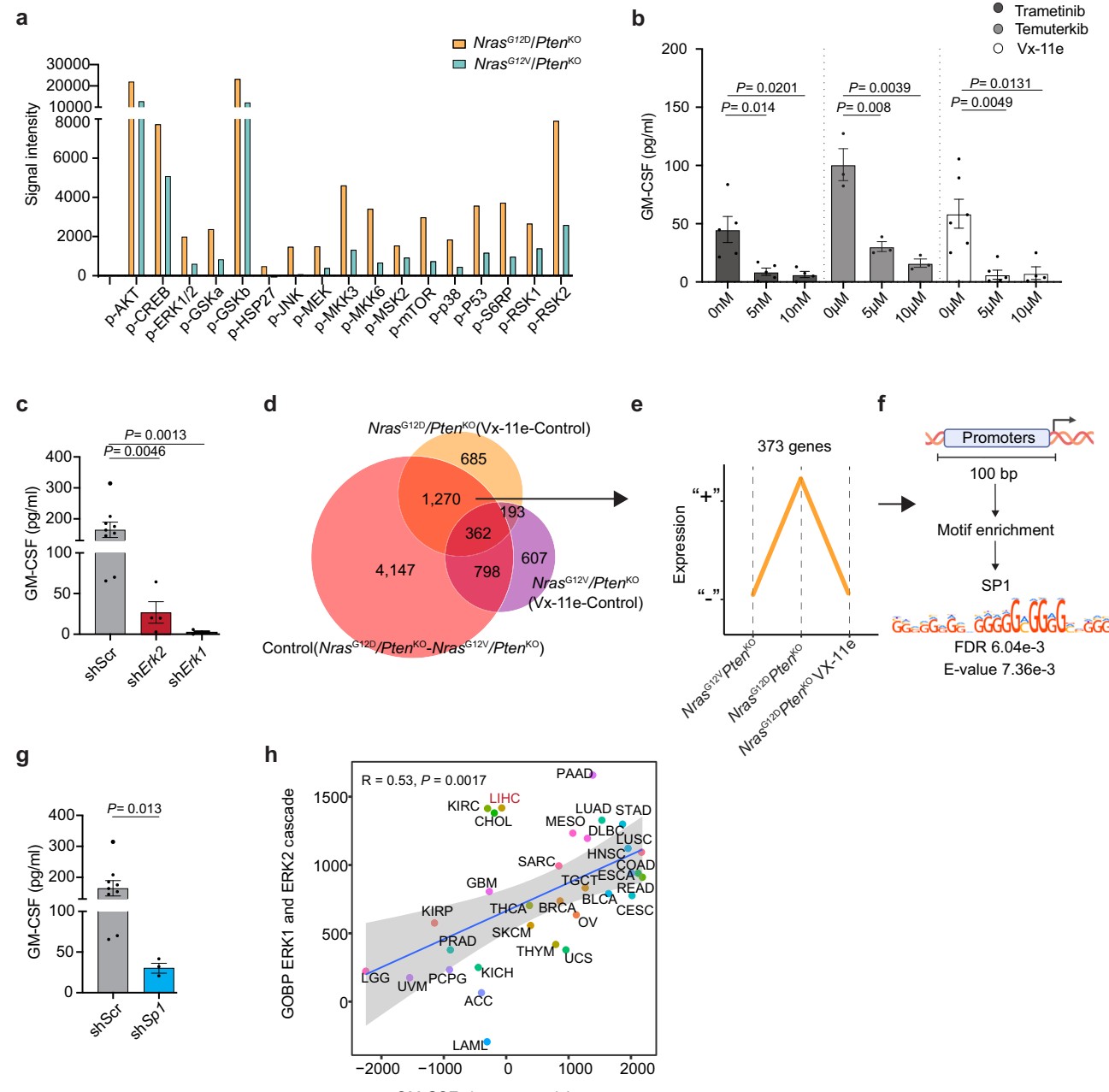

**Fig. 6 | GM-CSF expression is regulated through activation of the ERK1/2 pathway and SP1 transcription factor. a** Barplot depicting the expression levels (signal intensity) of RAS/MAPK-associated phospho-proteins in $Nras^{G12D}/Pten^{KO}$ and $Nras^{G12V}/Pten^{KO}$ HCC cell lysates ($n = 1$ per cell line). **b** Barplot depicting GM-CSF protein levels quantified in $Nras^{G12D}/Pten^{KO}$ cancer cell conditioned media after 24 h of treatment with Trametinib (MEK1/2 inhibitor; 0 nM $n = 5$, 5 nM $n = 5$, 10 nM $n = 4$), Temuterkib (ERK1/2 inhibitor; 0 µM $n = 3$, 5 µM $n = 3$, 10 µM $n = 3$) and Vx-11e (ERK2 inhibitor; 0 µM $n = 6$, 5 µM $n = 5$, 10 µM $n = 4$) at the indicated drug concentrations. **c** Barplots depicting the GM-CSF protein level quantified in the supernatant of scramble (sh$Scr$, $n = 9$), sh$Erk2$ ($n = 4$), and sh$Erk1$ ($n = 4$) $Nras^{G12D}/Pten^{KO}$ HCC cell lines. **d** Venn diagram depicting the overlap of DEG between $Nras^{G12D}/Pten^{KO}$ and $Nras^{G12V}/Pten^{KO}$ cancer cells treated or not with Vx-11e ($n = 3$ per cell line per condition) (Supplementary Data 9). **e** Graphical representation of the gene expression pattern (y-axis) across conditions (x-axis) that follow $Csf2$ regulation (**d**; 1270 overlapping DEG) shown (**d**). "+" indicates up-regulated genes and "-" indicates down-regulated genes. **f** Motif enrichment analysis in promoters from the 373 genes, identifying SP1 as a top candidate (Supplementary Data 9). **g** Barplots depicting the GM-CSF protein levels quantified in the supernatant of sh$Scr$ ($n = 9$, shown in (**c**)) and sh$Sp1$ ($n = 3$) $Nras^{G12D}/Pten^{KO}$ HCC cell lines. **h** Scatterplot depicting the correlation between the enrichment of ERK signaling pathway (y-axis) and GM-CSF signature[89] (x-axis) for each TCGA cancer type. Correlation analyses were performed on the median pathway enrichment score of all patients per cancer type. Graph shows mean ± SEM (**b**, **c**, **g**). Statistical significance was determined by two-sided Student's $t$-test (**b**, **c**, **g**), two-sided Fisher's exact test, followed by Benjamini-Hochberg multiple test correction (**f**) and two-sided test for association between paired samples, using Pearson's product moment correlation coefficient (**h**). The shading represents 95% confidence interval (**h**). Source data are provided as a Source Data file.

low $Ly6c1$, $Ly6c2$ and $Adgre1$ (F4/80) levels, and distinct gene expression patterns of $H2$-$ab1$, $H2$-$eb1$ and $Itgax$ (CD11c) (Supplementary Fig. 7i). Moreover, low expression of CD115 (CSF1R) and CX3CR1 together with high CD11b levels in Ly6C$^{low}$ myeloid cells (Fig. 7g, h)

further supports their immature macrophage state[97]. Overall, our data indicates that the Ly6C$^{low}$ myeloid cell pool is comprised of immature DC-like and macrophage-like monocytic cells that bear important pro-tumorigenic functions in $Nras^{G12D}/Pten^{KO}$ HCC. Indeed, loss of this

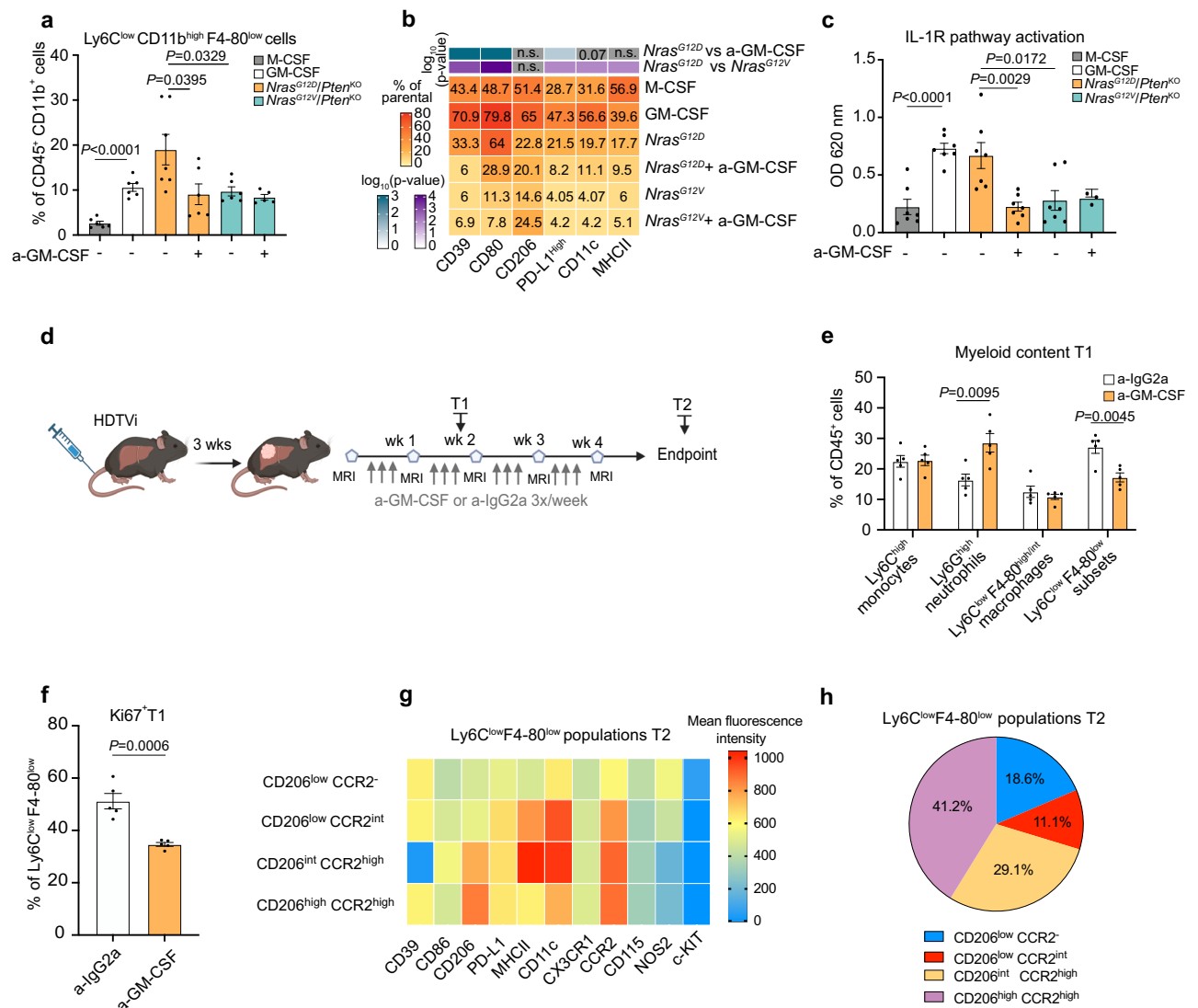

**Fig. 7 | GM-CSF blockade curbs monocyte-derived inflammatory Ly6C^low cell accumulation in the Nras^G12D/Pten^KO TME. a, b** Percentage of Ly6C^lowF4-80^low cells relative to total myeloid cells (a) obtained from bone marrow (BM) cells differentiated in either recombinant M-CSF (n = 6) or GM-CSF (n = 6), or in conditioned media (CM) prepared from distinct HCC cell lines, with or without GM-CSF neutralizing antibody (a-GM-CSF) (Nras^G12D/Pten^KO n = 7, Nras^G12D/Pten^KO + a-GM-CSF n = 6, Nras^G12V/Pten^KO n = 6, Nras^G12V/Pten^KO + a-GM-CSF n = 5) and (**b**) analyzed for the indicated phenotypic markers relative to the total Ly6C^low F4-80^low population. **c** IL-1R signaling activity in HEK reporter cells exposed to CM from BM cells differentiated with either recombinant M-CSF (n = 7), GM-CSF (n = 7), or to CM from Nras^G12D/Pten^KO or Nras^G12V/Pten^KO cell lines, in presence or not of a-GM-CSF (Nras^G12D/Pten^KO n = 7, Nras^G12D/Pten^KO + a-GM-CSF n = 7, Nras^G12V/Pten^KO n = 7, Nras^G12V/Pten^KO + a-GM-CSF n = 3). **d** Experimental design: mice were HDTV-injected to induce Nras^G12D/Pten^KO HCC, monitored by weekly MRI starting from 3 weeks post-injection, and enrolled into treatments with a-IgG2a (12.5 mg/kg three times per week) or a-GM-CSF (12.5 mg/kg three times per week) for 2 weeks (time point T1) or until end-stage (time point T2) for flow cytometry analyses. **e** Percentage of intratumoral Ly6C^high monocytes, Ly6G^high neutrophils, Ly6C^lowF4-80^high/int macrophages and Ly6C^lowF4-80^low subsets relative to total CD45^+ leukocytes in HDTVi-induced Nras^G12D/Pten^KO HCC-bearing mice 2 weeks post treatment (T1) with a-IgG2a (n = 5) or a-GM-CSF (n = 5). **f** Percentage of Ki67^+ intratumoral Ly6C^lowF4-80^low cells in HDTVi-induced Nras^G12D/Pten^KO HCC upon 2 weeks of treatment (T1) with a-IgG2a (n = 5) or a-GM-CSF (n = 5). **g** Mean fluorescence intensity of the depicted markers in four different Ly6C^lowF4-80^low subsets (CD206^lowCCR2^-, CD206^lowCCR2^+, CD206^+CCR2^high, and CD206^highCCR2^high) identified by FlowSOM analysis performed on HDTVi-induced Nras^G12D/Pten^KO end-stage (T2) HCCs (n = 3). **h** Proportions of Ly6C^lowF4-80^low subsets (CD206^lowCCR2^-, CD206^lowCCR2^+, CD206^intCCR2^high, and CD206^highCCR2^high) identified by FlowSOM analysis performed on HDTVi-induced Nras^G12D/Pten^KO end-stage (T2) HCCs (n = 3). Graphs show mean ± SEM (**a, c, e, f**) and median (**b**). Statistical significance was determined by unpaired two-sided Student's t-test in (**a–c, e, f**). See gating strategy in Supplementary Fig. 7b (a) and 9j (e-f). Source data are provided as a Source Data file.

myeloid cell subset upon GM-CSF blockade correlated with a significant decrease in tumor growth (Supplementary Fig. 7j), underscoring this treatment as a potential therapeutic approach in Nras^G12D/Pten^KO HCC.

**GM-CSF blockade curbs Nras^G12D/Pten^KO HCC outgrowth and cooperates with VEGF inhibition to prolong animal survival**

We next evaluated whether GM-CSF blockade would extend HCC-bearing mice survival. In light of the low penetrance of HDTV-induced

Nras^G12D/Pten^KO tumors (Supplementary Fig. 1b), we developed liver orthotopic HCC models by injecting Nras^G12D/Pten^KO cancer cells in WT C57/Bl6 mice (Supplementary Fig. 8a) (herein referred to as the liver orthotopic injection: LOI-model). Nras^G12D/Pten^KO LOI tumors exhibited comparable features to HDTVi-induced HCCs, with marked fibrosis (Supplementary Fig. 8b), high GM-CSF levels (Supplementary Fig. 8c), abundant myeloid cell content (Supplementary Fig. 8d) and comparable proportions of myeloid cell subsets within their TME (Supplementary Fig. 8e). GM-CSF blockade reduced the expression of GM-CSF

downstream genes *Ccl6* and *Ccl17* (Supplementary Fig. 8f, g), asserting treatment efficacy, and substantially curbed tumor growth (Supplementary Fig. 8h), leading to a significant increase in animal survival (Fig. 8a). Importantly, we observed that monocyte-derived Ly6C$^{low}$ cell abundance and proliferation were hindered in LOI tumors following a-GM-CSF treatment (Supplementary Fig. 8i, j), in line with prior in vitro (Fig. 7a) and in vivo results (Fig. 7e and Supplementary Fig. 7e).

We next compared the therapeutic efficacy of GM-CSF blockade to that of the recently approved standard-of-care (SOC) HCC therapy comprising dual inhibition of VEGF and PD-L1. Interestingly, a-GM-CSF and SOC prolonged animal survival to a similar extent (Fig. 8b), highlighting the advantage of a-GM-CSF as a stand-alone treatment regimen. Next, we interrogated whether combining GM-CSF blockade with a-VEGF and/or a-PD-L1 would increase therapeutic response, as *Vegfa*/ VEGF is strongly expressed in the *Nras*$^{G12D}$/*Pten*$^{KO}$ TME (Fig. 4f and Fig. 5c), while PD-L1 expression is heightened in *Nras*$^{G12D}$/*Pten*$^{KO}$ myeloid cells (Supplementary Fig. 3b). Strikingly, the abundance - but not the overall activation profile - of Ly6C$^{low}$ cells was decreased in all

treatment groups that displayed improved survival benefit compared to IgG-treated *Nras*$^{G12D}$/*Pten*$^{KO}$ tumors (Supplementary Fig. 9a, b). No significant changes were observed in the lymphoid compartment upon GM-CSF blockade, indicating a lack of adaptive T cell response (Supplementary Fig. 9c). In line with this result, incorporating PD-L1 blockade to GM-CSF neutralization did not enhance therapeutic response (Fig. 8b). Contrastingly, dual GM-CSF and VEGF blockade significantly extended animal survival when compared to anti-GM-CSF monotherapy or SOC (Fig. 8b). Mechanistically, the anti-tumorigenic effect of GM-CSF blockade involved an increase in intratumoral cleaved caspase 3 (CC3) (Fig. 8c, d) and decreased proliferation of parenchymal, non-immune cells (Supplementary Fig. 9d), the latter enforced to a further extent in the context of dual GM-CSF and VEGF blockade. This combination led to a significant increase in the presence of intratumoral necrotic areas compared to a-GM-CSF-treated *Nras*$^{G12D}$/*Pten*$^{KO}$ tumors (Fig. 8e). No visible changes in tumor vasculature or vessel density were observed (Supplementary Fig. 9e–g), suggesting a mechanism largely independent of the anti-angiogenic

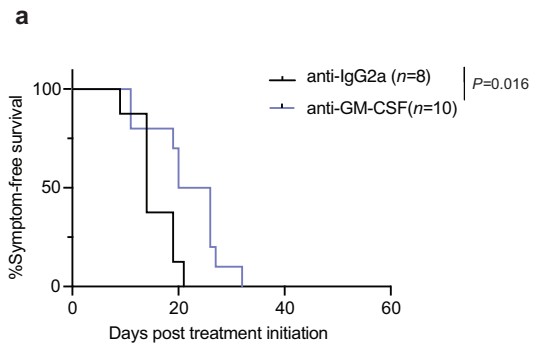

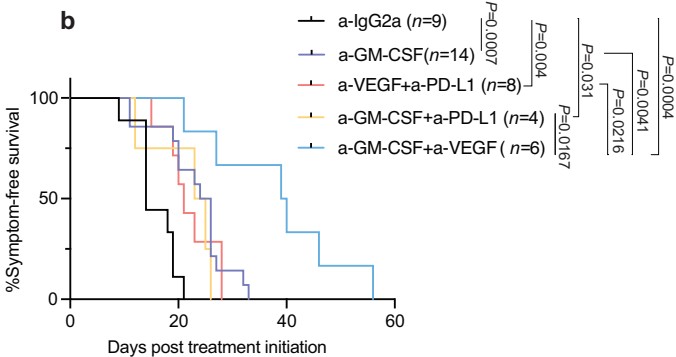

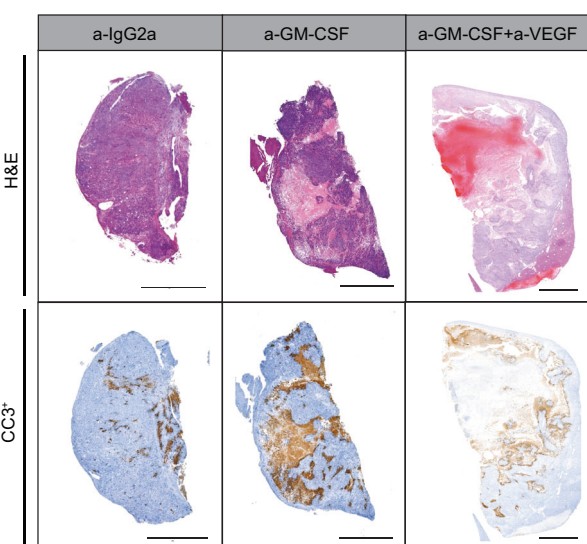

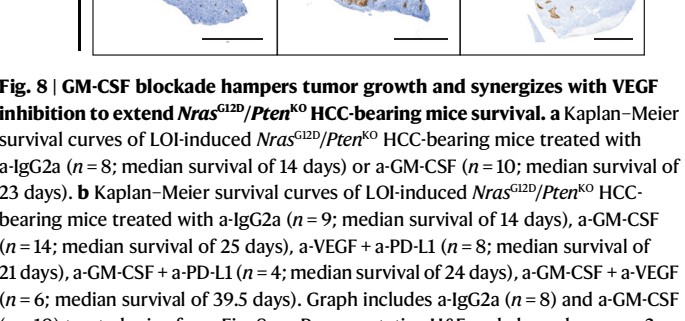

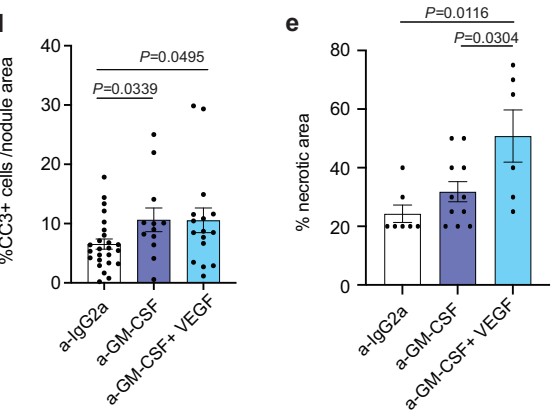

**Fig. 8 | GM-CSF blockade hampers tumor growth and synergizes with VEGF inhibition to extend *Nras*$^{G12D}$/*Pten*$^{KO}$ HCC-bearing mice survival. a** Kaplan–Meier survival curves of LOI-induced *Nras*$^{G12D}$/*Pten*$^{KO}$ HCC-bearing mice treated with a-IgG2a (n = 8; median survival of 14 days) or a-GM-CSF (n = 10; median survival of 23 days). **b** Kaplan–Meier survival curves of LOI-induced *Nras*$^{G12D}$/*Pten*$^{KO}$ HCC-bearing mice treated with a-IgG2a (n = 9; median survival of 14 days), a-GM-CSF (n = 14; median survival of 25 days), a-VEGF + a-PD-L1 (n = 8; median survival of 21 days), a-GM-CSF + a-PD-L1 (n = 4; median survival of 24 days), a-GM-CSF + a-VEGF (n = 6; median survival of 39.5 days). Graph includes a-IgG2a (n = 8) and a-GM-CSF (n = 10) treated mice from Fig. 8a. **c** Representative H&E and cleaved caspase 3 (CC3$^+$) IHC staining performed on liver sections from end-stage LOI-induced

*Nras*$^{G12D}$/*Pten*$^{KO}$ HCC-bearing mice treated with a-IgG2a, a-GM-CSF or a-GM-CSF + a-VEGF. (Scale bars = 2 mm; representative CC3$^+$ for data shown in d and representative H&E for data shown in (**e**)). **d** Barplot depicting the percentage of cleaved caspase 3 (CC3$^+$) positive cells per area in tumor nodules (end-stage) isolated from LOI-induced *Nras*$^{G12D}$/*Pten*$^{KO}$ HCC-bearing mice treated with IgG2a (n = 9), a-GM-CSF (n = 5) or a-GM-CSF + a-VEGF (n = 6). **e** Barplot depicting the percentage of necrotic area in tumor section (end-stage) from LOI-induced *Nras*$^{G12D}$/*Pten*$^{KO}$ HCC-bearing mice treated with IgG2a (n = 7 mice), a-GM-CSF (n = 11 mice) or a-GM-CSF + a-VEGF (n = 6 mice). Graph shows mean ± SEM (**d**, **e**). Statistical significance was determined by log-rank test (**a**, **b**) and unpaired two-sided Student's t-test (**d**, **e**). Source data are provided as a Source Data file.

effect of VEGF blockade. As $Nras^{G12D}/Pten^{KO}$ tumor cells express higher levels of $Flt1$ (encoding VEGFR-1) compared to its $Nras^{G12V}/Pten^{KO}$ counterpart (Supplementary Fig. 9h), we propose that the blockade of the VEGF-VEGFR-1 signaling axis in cancer cells is associated with lower proliferation and increased tumor necrosis, thereby synergizing with the myeloid-centric effects of GM-CSF neutralization in a two-drug combination regimen, similarly to the currently approved standard of care.

Altogether, these results suggest that a MAPK-ERK1/2-SP1-GM-CSF signaling node underlies several aspects of cancer cell-intrinsic pro-tumorigenic features, through increasing the abundance of monocyte-derived Ly6C$^{low}$ cells in the $Nras^{G12D}/Pten^{KO}$ HCC TME and promoting a heightened inflammatory and immunosuppressive phenotype in this immature myeloid cell population. Interfering with these processes by blocking GM-CSF enforces a therapeutic vulnerability in the HCC TME that synergizes with the clinically-approved VEGF inhibition (Supplementary Fig. 9i).

## Discussion

Faithfully recapitulating the HCC multilayered heterogeneity is essential to identify therapeutic approaches that can tackle the unmet clinical need of this disease. Here, we generated a collection of advanced precision models of HCC, each harboring genetic driver mutations altering clinically-relevant oncogenic signaling pathways that closely mimicked human HCC pathology and heterogeneity[4,75]. Notably, the myeloid-dominated $Nras^{G12D}/Pten^{KO}$- and $Myc^{OE}$-driven HCCs ($Myc^{OE}/Trp53^{KO}$, $Myc^{OE}/Pten^{KO}$) correlated with the aggressive HCC proliferation subtypes, supporting the notion that myeloid cell recruitment enforces tumor aggressiveness[98]. $Myc^{OE}$-driven models consistently clustered together, underlying the dominant role of MYC in shaping both cancer cell-intrinsic and extrinsic features of HCC fueling an immune-desert and immunosuppressive TME in several tumor types[17,18,21,26,56,99,100].

While previous reports highlighted tissue-dependency and isoform specificity related to RAS signaling[50,101], our study address the extrinsic effects of differential Ras/MAPK signaling pathway activation in cancer. Interestingly, G12D and G12V point mutations hold different enzymatic activity levels[102], which might explain the differential MAPK signal intensity observed in the two $Nras$-driven models. Although $Ras$ mutations are rare in HCC patients (~1%)[103], the Ras/MAPK pathway is overactivated in over 50% of them[46], highlighting the translational potential of these findings. Herein, we revealed that varying degrees of Ras-MAPK signaling pathway intensity control unique biological and clinical behaviors in HCCs, with $Nras^{G12D}$-driven HCC exhibiting a myeloid-enriched and fibrotic TME, and $Nras^{G12V}$ eliciting a T cell-inflamed and lipid-enriched tumors, resembling NASH-like features. Of note, cancer cell-intrinsic lipid metabolic reprograming in $Nras^{G12V}/Pten^{KO}$ HCC may be therapeutically exploitable in future studies. Our results emphasize the relevance of investigating the activation states pertaining to a specific oncogenic signaling pathway to unravel the molecular bases of inter-patient heterogeneity.

The MAPK-enriched, $Nras^{G12D}/Pten^{KO}$-driven HCCs displayed a myeloid-rich TME, uniquely presenting a mixed inflammatory and immunosuppressive profile, thus highlighting the distinctive and yet under-appreciated immune-modulation capacity of the MAPK/ERK pathway hyperactivation through $Nras^{G12D}$ point mutation. Indeed, cooperation of $Kras^{G12D}$ and $Myc$ oncogene was previously shown to fuel invasive lung adenocarcinoma immune-suppressed stroma[56] while $Nras^{G12V}$-induced senescent cells elicited a T cell response and modulated myeloid cell recruitment and maturity in the hepatic environment[35,55]. Complementing these reports, we unbiasedly identified GM-CSF as a key regulator of local myeloid cell education exclusively in $Nras^{G12D}$-associated HCC, in a ERK1/2-SP1 dependent manner. In line with previous reports that identified GM-CSF as a pilot signal in $Kras^{G12D}$-driven PDAC and cholangiocarcinoma[86,104], we

provide evidence of an analogous regulatory mechanism pertaining to distinct Ras isoforms in HCC. Moreover, our study unveils ERK1/2 and the downstream SP1 transcription factor as regulators of GM-CSF transactivation in HCC[105–107]. Hence, these findings shed light into alternative therapeutic targets exploitable in GM-CSF-enriched solid cancers.

Remarkably, while GM-CSF is known as a central mediator of myeloid cell function and differentiation[87–89], we uncovered the intricate connections between the GM-CSF and IL-1 signaling nodes specifying monocytic cell phenotype. We exposed a GM-CSF-driven local expansion of proinflammatory/immunosuppressive CCR2$^+$ immature Ly6C$^{low}$ myeloid cells within the $Nras^{G12D}/Pten^{KO}$ TME. The exact nomenclature for such subset is currently lacking, as these cells display characteristics associated with both macrophages and DCs, a confounding phenotype previously reported in progenitor cells exposed to GM-CSF in vitro[90]. Importantly, the anti-tumorigenic effects of GM-CSF neutralization involved the reduction and re-education of monocyte-derived Ly6C$^{low}$ cells and the induction of cancer cell death in a T cell-independent manner. These results contrast with previous reports in $Kras^{G12D}$-driven tumors[86,104] in which T cells are central to GM-CSF blockade anti-tumor effect, suggesting the context dependency of this treatment. As inflammation is known to sustain cancer cell propagation and survival[81], we propose that GM-CSF blockade hinders a myeloid-centric inflammatory cascade upheld by Ly6C$^{low}$ immature cells. Nevertheless, acquired tumoricidal capacity in myeloid cells could also be hypothesized, as recently reported in metastatic disease following macrophage reprogramming[108]. Overall, our findings further expand the current appreciation of non-cell autonomous effects held by specific signaling pathways on the TME dynamics and therapy response. Indeed, recent generation of HCC somatic models were used to tailor anti-cancer therapies to HCC mutational background[26]. Here, we therapeutically exploited the myeloid-driven pro-tumorigenic effects orchestrated by GM-CSF to rewire the $Nras^{G12D}$ TME into sustaining cancer cell survival and proliferation, a process further hindered upon combination with the clinically-approved VEGF inhibition, which led to increased tumor necrosis. As GM-CSF signature correlates with both poor patient prognosis and the $Nras^{G12D}/Pten^{KO}$-specific transcriptome, we advance the potential of GM-CSF neutralization as a distinctive myeloid cell centric immunomodulation strategy in HCC, which can be further enhanced in combination with VEGF blockade standard of care. As this combination sufficiently promoted a prolonged survival in $Nras^{G12D}/Pten^{KO}$ HCC-bearing mice beyond the standard of care treatment (combined VEGF and PD-L1 blockade), we believe this two-drug combination could elicit a satisfactory therapeutic response in this subset of patients that correlate with the $Nras^{G12D}/Pten^{KO}$-specific transcriptome, while limiting the possible side effects resulting from combining the standard of care with GM-CSF blockade in a three-drug regimen.

Recent changes in the treatment guidelines for advanced HCC patients have reinforced the importance of harnessing the TME as a successful therapeutic strategy[10]. Our study expands the understanding of HCC inter-tumor heterogeneity by revealing how cancer cell-intrinsic oncogenic pathways shape clinicopathological and TME profiles. Altogether, these findings set the basis for the design of personalized immune intervention strategies tailored to cancer cell-intrinsic features in HCC patients and expose alternative treatment approaches for this disease.

## Methods

### HCC model generation and treatment

All animal experiments were reviewed and approved by the Animal Ethics Committee of the Netherlands Cancer Institute and performed in accordance with institutional, national and European guidelines for Animal Care and Use. All HCC mouse models were generated in C57BL/6J background from 6–8 weeks old females (Janvier laboratories). In

the interest of ensuring reproducibility of tumor kinetics, penetrance and growth, the animal experiments were conducted using only one gender.

HCC somatic mouse models were generated using hydrodynamic tail vein injections (HDTVi)[47,109]. Briefly, a volume equivalent to 10% of mouse body weight of sterile 0.9% NaCl saline solution containing plasmid mixtures was injected into the mouse lateral tail vein in 7–10 s. A total of 25 μg of DNA mixture was injected per mouse and prepared as followed: 10 μg of transposon vector, 10 μg of CrispR/Cas9 vector, and 5 μg of Sleeping beauty (SB) transposase encoding vector (unless indicated). The *CMV-SB13*, the *pT3-EF1a-Myc* and the *pT3-EF1-Nras*[G12D]-*GFP* vectors were a kind gift from Dr Scott Lowe (Memorial Sloan Kettering Cancer Center, New York, USA). The *pT/CaggsNras*[G12V]-*IRES-Luc* vector[110] was kindly provided by Lars Zender (University of Tuebingen, Tuebingen, Germany), the *pX330-Trp53* (Addgene 59910), the *pX330-Pten* (Addgene 59909), were previously validated and are publicly available. Mice were followed with weekly and bi-weekly MRI starting 1-week post-HDTVi to measure the longitudinal progression of tumor volumes[109,111]. Animals were sacrificed when symptomatic and/ or when tumor volume reached a total volume ≥2 cm³ in survival curves and end-stage analyses as approved as humane endpoint by the Animal Ethics Committee of the Netherlands Cancer Institute. For time-point analysis, HCC-bearing mice were sacrificed 1–2 weeks (*Myc*[OE]-driven models) and 2–3 weeks (*Nras*-driven models) post-tumor development, corresponding to a tumor volume ≥600 mm³. Control mice underwent HDTVi of a DNA mixture prepared as described above with scramble empty vectors publicly available: *pT3-EF1-Neo-GFP* (Addgene 69134), and the *pX330-CBh-hSpCas9* (Addgene 42230) and were sacrificed 4 weeks (as referred to 4-weeks control) or 15 weeks (as referred to 15-weeks control) later.

For the generation of liver orthotopic (LOI) HCC mouse models, intrahepatic injection of HCC cell lines - isolated from HDTVi-driven tumors - was carried out according to previously established protocols[112]. Briefly, $1.0 \times 10^5$ *Nras*[G12D]/*Pten*[KO] cancer cells were resuspended in 5 μl of serum-free DMEM medium supplemented with 25% Matrigel (Corning, cat.no. 356230). Through an 8-mm midline, central incision, cells were slowly injected in the left-medial and/or left lateral liver lob using a microfine insulin syringe. Mice were followed with weekly MRI starting 7 days post-injection to measure tumor progression. For the a-GM-CSF, a-PDL-1, a-VEGF preclinical trials, when tumors were first visible by MRI, tumor size-matched animals (HDTVi- or LOI-generated) were randomized in the indicated treatment groups. Mice were treated three times a week intraperitoneally with a-IgG2A control (12.5 mg/kg BioXcell, BE0089), a-GM-CSF (12.5 mg/kg BioXcell, BE0259), a-PD-L1 (10 mg/kg BioXcell, BE0361), a-VEGF (10 mg/kg B20S, kindly gifted by Dr Iacovos Michael). HDTVi- *Nras*[G12D]/*Pten*[KO] mice were treated until sacrificed at 2 weeks post treatment (timepoint: T1) or at humane endpoint (timepoint:T2). LOI- *Nras*[G12D]/*Pten*[KO] mice were treated until sacrificed, either upon exhibiting symptoms (significant weight loss of ≥15% of body weight within 2 days, or a 20% weight loss since the start of experiment, along with severe circulation and breathing problems or aberrant behavior/movements) or when tumor volume reached a total volume ≥2 cm³ in survival curves and end-stage analyses as approved by the Animal Ethics Committee of the Netherlands Cancer Institute.

### Tissue microarray analyses of HCC patient samples
**Patient information.** An independent cohort of 488 HCC patients who underwent primary curative resection between 2006 and 2010 was enrolled. The resection procedure and postoperative surveillance for recurrence were performed as described previously in refs. [69,113]. The Institutional Review Board of Sun Yat-sen University Cancer Center approved this study and all samples were anonymously coded in accordance with local ethical guidelines, as stipulated by the Declaration of Helsinki, with written informed consent obtained from all participants. The clinical characteristics of patient are summarized in Supplementary Data 4.

### IHC and image acquisition
Tissue microarrays of HCC samples were cut into 4 μm sections, and then processed for IHC according to our previous reports[69,113]. Primary antibodies against CD11b (ab133357;Abcam; 1:4000), CD15 (ZM-0037; ZSBio; 1:100), S100A9 (34425; Cell Signaling Technology (CST); 1:400), CD204 (KT022;Transgenic; 1:100) were used to determine myeloid cell subsets, p-ERK1/2 (ab214036;Abcam; 1:500) and c-Myc (ab32072, Abcam; 1:100) were used to evaluate the activation of signaling pathways. IHC-stained slides were scanned at ×20 magnification by digital pathology slide scanner (KFBIO).

### Image analysis
Cell numbers of myeloid subsets were estimated with object module of InForm Tissue Analysis Software (AKOYA). Myeloid response score was determined as described previously in refs. [69,113]. To define p-ERK1/2[+] and c-MYC[+] tumors, samples displaying unequivocal nuclei staining were classified as positive by 2 independent observers who were blinded to the clinical outcome.

### Cell lines, culture conditions and ex vivo experiment
*Myc*[OE]/*Trp53*[KO], *Myc*[OE]/*Pten*[KO], *Nras*[G12D]/*Pten*[KO], and *Nras*[G12V]/*Pten*[KO] HCC cell lines were generated and isolated from end-stage tumor-bearing mice of each distinct HCC mouse model as previously described in ref. [109]. Cells were grown on Collagen Type I rat tail (Corning, cat.no. 354236) pre-coated flasks/dishes and cultured using DMEM (Gibco, cat. no. 61965059) supplemented with 10% FCS, 1x penicillin/streptomycin (Roche) (referred to as complete medium).

AML12 cells were provided as a kind gift from Urszula Hibner (Institut de Genetique Moleculaire de Montpellier, France) and cultured with DMEM-F12+GlutaMax supplemented with 10% FBS (Capricorn, cat.no. FBS-12A), 1% penicillin/streptomycin (Roche) and 1% insulin-Transferrin-Selenium (ITS) (Gibco) (referred to as complete F12 medium)[114].

HEK-Blue[TM] cells (Invivogen cat. code hkb-il1r) were cultured in complete medium supplemented with 100 μg/mL Normocin according to the manufacturer's instructions.

All cell lines were cultured at 37 °C and 5% $CO_2$ and routinely tested for mycoplasma contamination using a MycoAlert® mycoplasma detection kit (Lonza, cat: LT07-218). Only mycoplasma-negative cells were used. The purchased cell lines were not authenticated.

### Generation of shRNA cell lines
The lentiviral PLKO.1-puro vectors containing a short hairpin RNA (shRNA) targeting *Erk1* (5'- AGGACCTTAATTGCATCATTA-3'), *Erk2* (5' GCTCTGGATTTACTGGATAAA-3') and *Sp1* (5' CCTTCACAACTCAAG CTATTT-3') were obtained from the TRC library. Control (scrambled) vector was available through purchase (Addgene #136035). A total of $2 \times 10^6$ HEK 293 T cells (kindly gifted by Prof. Karin de Visser's lab) were seeded and after 24 h transfected with 1.5 μg of pLKO.1 vector encoding shRNAs, 1 μg of pPAX packaging vector and 1 μg of VSV-G envelope vector using FuGENE® HD Transfection Reagent (Promega, cat.no. E2311), according to the manufacturer's instructions. 24 h after transfection, supernatants were collected, filtered (0.45-mm pore size filter; Millipore), and added to *Nras*[G12D]/*Pten*[KO] cells to be transduced for 16 h. After 48 h, transfected cells were selected with puromycin (2.5 μg/ml) for 4 days.

### Generation of conditioned media
To generate conditioned media (CM), $5 \times 10^6$ HCC cell lines or AML12 cells were seeded in complete medium and/or complete F12 medium, respectively. After 24 h, medium was replaced with 0% FBS DMEM to

generate CM. After 24 h, CM was collected and centrifuged at $1000 \times g$ for 5 min to remove cell debris and stored at −80 °C and subsequently used for bone marrow cell differentiation and cytokine array.

## Bone marrow cell differentiation

Bone marrow (BM) cells were freshly isolated from wild-type mice as previously described in ref. 31. Briefly, both femurs and tibias were flushed in complete medium using a 23G needle. The cell suspension was filtered through a 100 µm cell strainer (Millipore) and cultured in 10 ml Teflon bag (OriGen PermaLife) with either control conditions or CM generated from HCC cell lines (as explained above) supplemented with 2% FBS, in the presence or absence of 5 µg/ml of a-GM-CSF (BioXcell, cat.no. BE0259). M-CSF and GM-CSF differentiated BM cells were cultured with complete medium +10 ng/ml recombinant mouse M-CSF (Biolegend, cat. no. 576408) or +20 ng/ml recombinant GM-CSF (Peprotech, cat. no. 315−03), respectively. Cells were maintained in culture for a total of 5 days and medium was refreshed every 2 days. On day 5, the cell suspension was harvested and centrifuged at $300 \times g$ for 5 min. Supernatants were used for HEK-Blue™ IL-1R assay. Cell pellets containing differentiated BM cells were used for RNA isolation and flow cytometry analyses.

## HEK-Blue IL-1R assay

IL-1R signaling activity was measured using HEK-Blue™ IL-1R cells (Invivogen, cat. code hkb-il1r) following the manufacturer's instructions. Briefly, differentiated BM cell supernatant was centrifuged at $1000 \times g$ for 5 min to remove cell debris. 200 µl of 10× diluted-supernatant was added to $5 \times 10^4$ HEK-Blue™ IL-1R cells seeded in 96-well plate. Complete medium supplemented or not with recombinant IL-1β (25 ng/ml) (Abcam, cat.no. ab259421) was used as a positive and negative control, respectively. The next day, 20 µl of HEK cell supernatant was transferred to a flat-bottom 96-well plate containing 180 µl of QuantiBlue (Invivogen, cat. code rep-qbs) detection reagent and incubated at 37 °C for 30 min. IL-1R activity was determine with a spectrophotometer (Tecan) at 620 nm emission. Negative background was subtracted from the raw values.

## MAPK and PI3K/mTOR pathway inhibition

$5 \times 10^5$ HCC cell lines were seeded into 6-well plates with complete medium. The following day, cells were cultured in 0% FBS-DMEM for 24 h with: Trametinib, MK2206, AZD8055, SP600125, Vx-11e and Temuterkib (S2673, S1078, S1555, S1460, S7709, S8534, Selleck Chemicals). Supernatants were collected to determine the secreted GM-CSF levels. Cell pellets were used for protein or RNA isolation.

Cell growth was determined by IncuCyte ZOOM (Essen BioScience) assays as previously described in ref. 111. Briefly, $2 \times 10^4$ $Nras^{G12D}/Pten^{KO}$ HCC cells were seeded in 48-well plates and treated with the MAPK and PI3K/mTOR pathway drugs mentioned above in DMEM-FCS 0%. Cells were imaged every 4 h and phase-contrast images were analyzed to determine the relative cell growth based on cell confluency.

## RNA isolation, cDNA synthesis, and RT-qPCR

RNA extracted from snap frozen intermediate and end-stage bulk tumors and from 4- to 15-week control HDTVi livers (~5 mg), from bulk tumor FACS-sorted myeloid cell populations, or from primary HCC cell lines was isolated using TRIzol (Thermo Fisher) according to the manufacturer's instructions. RNA from educated BM cells was isolated using RNAeasy kit (Qiagen, cat. no. 74104).

For cDNA synthesis, the High-Capacity cDNA Reverse Transcriptase Kit (Thermo Fisher, cat. no. 4368814) was used with 500 ng of RNA. The following Taqman probes (Thermo Fisher) were used for qPCR: *Ubc* (Mm01201237_m1), *Il6* (Mm00446190_m1), *Il1b* (Mm00434228_m1), *Il1a* (Mm00439620_m1), *Ccl6* (Mm01302419_m1), *Ccl17* (Mm00516136_m1), *Csf2* (Mm01290062_m1), *Nlrp3*

(Mm00840904_m1), *Arg1* (Mm00475988_m1) and *Sp1* (Mm00489039_m1). Relative expression was calculated after normalization to the housekeeping gene *Ubc* for each sample.

## Protein isolation

Proteins were isolated from HCC cell lines using RIPA Lysis buffer (Thermo Fisher, cat. no. 89900) supplemented with Halt™ Protease and Phosphatase Inhibitor Cocktail (Thermo Fisher). Snap frozen tumor samples (~5 mg) were lysed in cOmplete™ Lysis-M lysis buffer (Roche, cat. no. 4719956001) supplemented with protease and phosphatase[111]. Protein lysates were sonicated and protein concentration was determined using Pierce™ BCA Protein Assay Kit (Thermo Fisher, cat. no. 23225) for subsequent analyses.

## Western blot

Equal concentrations of proteins from total cell lysates (25–50 µg) were loaded on SDS-PAGE gels and transferred onto PVDF membranes. The membranes were blocked for 1 h with 5% milk PBS + 0,05% Tween (PBS-T) and incubated overnight at 4 °C with primary rabbit antibodies against p-AKT (4060S; CST; 1:1000 dilution), p-c-Jun (3270 S; CST; 1:1000 dilution), p-ERK1/2 (4370S; CST; 1:500 dilution), T-ERK1/2 (9102S; CST; 1:1000 dilution), p-S6RP (211S; CST; 1:1000 dilution), p-4EBP1 (9459S; CST; 1:1000 dilution), Ras-G12D (14429S; CST; 1:500 dilution), p-RSK-1 (9341S; CST; 1:1000 dilution), and vinculin (13901T; CST; 1:1000 dilution) Secondary conjugation was performed using anti-rabbit IgG, HRP-linked Antibody (CST, cat. no. 7074P2) for 1 h at room temperature, and proteins were detected with Signal Fire™ ECL Reagent (CST, cat. no. 6883P3) using BioRad ChemiDoc™ XRS+ System. Bands from western blots were quantified using Image Lab Software (BioRad).

## Caspase-1 assay

To measure caspase-1 activity, end-stage bulk tumor and 4- to 15-weeks control HDTVi liver protein lysates (50 µg) were analyzed using Fluorometric Caspase-1 Assay Kit (Abcam, cat. no. ab273268) according to the manufacturer's instructions. Signal was acquired using the TECAN plate reader at an emission of 505 nm and excitation of 400 nm. Fold change values were obtained after normalizing the samples by dividing each value to one control liver sample following background subtraction.

## Cytokine array

Measurement of cytokine levels were assessed in HCC cancer cell's CM, end-stage tumor bulk and 15-weeks control HDTVi liver lysates (1000 µg), and 5x diluted serum from end-stage HCC-bearing and 15-weeks control mice using the Proteome Profiler Mouse XL Cytokine Array (R&D systems, cat.no. ARY028) according to the manufacturer's instructions. Pixel density was quantified using ImageLab software (Biorad). Each membrane array was normalized to its own reference control after subtracting the background signal. Fold change values of pixel densities were obtained by using control samples (AML12 for cell lines, 15-weeks control mice for serum and bulk analyses) as the baseline value (set to 1). Measurements shown in Supplementary Fig. 5c, d, k represent $n = 1$ independent experiment each. Results of Supplementary Fig. 5c, d were used to validate transcriptional candidates that were differentially regulated at bulk tumor and cell line level in the *Nras*-driven models in Fig. 5c. The main target candidate, *Csf2*, was further validated by GM-CSF ELISA in various independent experiments in Fig. 5d.

## Multiplex assay

GM-CSF level (shown for this assay in Supplementary Fig. 5i) were assessed in end-stage HCC and 4- to 15-weeks control HDTVi liver protein lysates (350 µg) by Protavio Ltd with a 11-plex array, and analyzed according to the Luminex technology company's protocol.

## Phospho protein profiling

To measure the phosphorylated levels of RAS-associated effector proteins, *Nras*$^{G12D}$/*Pten*$^{KO}$ and *Nras*$^{G12V}$/*Pten*$^{KO}$ cell line protein lysates (1000 µg each) were analyzed using the Proteome Profiler™ Phospho-MAPK Array Kit (Raybiotech, cat.no. AAH-AMPK-1-2) according to manufacturer's instructions. Pixel intensity was quantified using ImageLab Software (Biorad). Each membrane array was normalized to their own reference controls after subtracting the background signal. Measurements shown in Fig. 6a represent independent experiment (*n* = 1 for each cell line). The results were further validated by western blot in Supplementary Fig. 6a, with *n* = 4 independent experimental repeats.

## GM-CSF ELISA assay

To quantify the levels of secreted GM-CSF, CM was generated as mentioned above and GM-CSF was quantified using the mouse GM-CSF uncoated ELISA Kit (Thermo Fisher, cat.no. #88-7334-22) in accordance with the protocol provided by the manufacturer. GM-CSF concentration was normalized to the protein concentration of the respective HCC cell line lysate, quantified with Pierce™ BCA Protein Assay Kit (Thermo Fisher).

## Tissue collection and processing

Mice were euthanized by carbon dioxide asphyxiation. Blood was collected by heart puncture and subsequently transcardially perfused with PBS until liver is cleared of blood. Harvested livers were macro-dissected and used for further analysis[109].

## Serum preparation

Blood was left to clot at room temperature for 15 min and then centrifuged at 2000 × *g* for 20 min at 4 °C. Serum was collected from the supernatant and stored at −80 °C until further use.

## ALT activity

To assess hepatocellular damage, ALT activity was measured with the Alanine Transaminase Activity Assay Kit (Abcam, colorimetric, cat.no. ab105134) using the serum (5 × diluted) of intermediate- and end-stage HCC-bearing and 4- to 15-weeks control mice according to the manufacturer's instructions. Signal was acquired using the TECAN plate reader at 570 nm emission and 37 °C in 5-minute intervals during a total scanning time of 90 min. Negative background was subtracted from the raw values.

## AFP ELISA assay

To quantify the levels of AFP at the systemic level, Mouse alpha Feto-protein ELISA kit (Abcam, colorimetric, cat. no. ab210969) was performed using the serum (5x diluted) of intermediate-stage HCC-bearing and 4- to 15-weeks control mice according to the manufacturer's instructions. Signal was acquired using the TECAN plate reader at 450 nm emission. Negative background was subtracted from the raw values.

## Tissue imaging and analysis

Murine liver specimens were obtained from HCC end-stage and control-15-weeks mice and processed for IHC and histochemistry (HC) analysis as previously described in ref. 109. Briefly, formalin-fixed, paraffin-embedded samples were sectioned at 4 µm and either probed with the indicated antibodies listed in Supplementary Data 10 or stained with Masson's trichrome dye for collagen fibers staining. Alternatively, samples were frozen down in Optimal Cutting Temperature (OCT) compound (Tissue-Tek), sectioned at 10 µm and stained with Oil red O dye for lipid staining.

All stained slides were digitally processed using the Aperio ScanScope (Aperio, Vista, CA) at a magnification of 20×. Immunohistochemical and histochemical staining were performed by the Animal Pathology facility at the Netherlands Cancer Institute.

Histopathological evaluation of HCC tissue samples was performed on H&E, ARG1, Masson's trichrome and Oil red O-stained slides by experienced liver pathologists (dr. Joanne Verheij, Amsterdam UMC and Dr. Ji-Ying Song, NKI-Avl). For the analysis of CC3 stained HCC tumor samples, nodule size and the nodule areas containing CC3+ cells were drawn by hand on HALO image-analysis software (Indica Labs) to quantify the percentage of CC3+ per nodule area.

## Flow cytometry

Macrodissected HCC nodules and control liver samples were dissociated as single-cell suspensions using the Liver Dissociation kit (Miltenyi Biotec) and the gentleMACS Octo Dissociator following the manufacturer's instructions. Samples were incubated with anti-CD16/CD32 antibody (BD Bioscience) and stained with the antibodies against surface markers following standard procedures. Samples were fixed with the eBioscience fixation and permeabilization kit (Invitrogen), and stained for intracellular markers (see Supplementary Data 11). All antibodies used for flow cytometry were titrated in a lot-dependent manner and are listed in Supplementary Data 11. All analyses were completed at the Flow Cytometry Core facility at the NKI. For tumor and blood samples from the genetically-distinct HCC mouse models, the following immune cells were identified. For myeloid cells: DCs (CD45$^+$ F4/80$^-$ CD11c$^{high}$MHCII$^{high}$), neutrophils (CD45$^+$ CD11b$^+$ Ly6C$^{int}$ Ly6G$^+$), Ly6C$^{high}$ monocytes (CD45$^+$ CD11b$^+$ Ly6C$^{high}$ Ly6G$^-$), Ly6C$^{low}$ monocytes (blood)/ subsets (Tumor) (CD45$^+$ CD11b$^+$ Ly6C$^{low}$ Ly6G$^-$), monocyte-derived macrophages (MDMs; CD45$^+$ CD11b$^+$ Ly6C$^{low}$ Ly6G$^-$ F4-80$^{int}$CD11b$^{high}$) (Supplementary Fig. 9j); for lymphoid cells: CD8$^+$ T cells (CD45$^+$ CD11b$^-$ NK1.1$^-$ CD19$^-$ CD3$^+$ CD8$^+$CD4$^-$), CD4$^+$ T cells (CD45$^+$ CD11b$^-$ NK1.1$^-$ CD19$^-$ CD3$^+$ CD8$^-$ CD4$^+$) (Supplementary Fig. 9k); For blood, intermediate- and end-stage tumor analyses a four-laser BD Fortessa instrument (BD Bioscience, BD FACSDiva software v 8.0.2) was used in Fig. 3 and Supplementary 3, 7c. For multidimensional data visualization and analyses in Fig. 7 and Supplementary 8, 9, a five-laser Aurora spectral flow cytometer (Cytek Biosciences) was used. Data was analyzed using the FlowJo v10 software. The Ly6C$^{low}$F4-80$^{low}$ populations obtained in the analyses was downsampled using the DownSample 3.3.1 plugin (FlowJo Exchange). The FlowSOM 3.0.18 and UMAP 3.1 plugins were used to analyze the Ly6C$^{low}$F4-80$^{low}$ populations from a concatenated dataset.

## Fluorescence-activated cell sorting

All sorting experiments were performed at the Flow Cytometry Core facility of the NKI with a BD FACSAria Fusion sorter (BD Bioscience) using a 100 µm nozzle.

For RT-qPCR gene expression analysis, macrodissected HCC nodules and 4- to 15-weeks control livers were dissociated in a single-cell suspension and stained with antibodies as listed in Supplementary Data 11. Viable cells (NIR$^{low}$) were sorted in 2% FBS-PBS based on CD45$^+$CD11b$^+$Ly6G$^-$Ly6C$^{high}$ and CD45$^+$CD11b$^+$Ly6C$^-$Ly6G$^-$F4-80$^{int}$ to isolate Ly6C$^{high}$ monocytes and MDMs, respectively. The sorted cells were centrifuged at 300 x g for 5 min at 4 °C, washed with PBS and stored at −80 °C in TRIzol (Thermofisher) for RNA isolation.

For transcriptomic analysis of the immune cell content, single viable cells (NIR$^{low}$) were sorted based on CD45$^+$ expression. 2% FBS-PBS sorted cells were centrifuged at 300 × *g* at 4 °C and the cell pellets were stored at −80 °C for RNA sequencing.

For single-cell RNA sequencing (scRNA-seq), three control mice (4 weeks post-HDTVi) were digested and pooled in a single-cell suspension. Samples were stained with live/dead staining using Zombie NIR (Biolegend) and CD45 and CD11b (see Supplementary Data 11). Single viable (NIR$^{low}$) cells were sorted based on CD45 expression (negative and positive) in 2% FBS-PBS. Sorted cells were washed with PBS with 0,04% BSA (PBS-BSA), and purified CD45$^+$ and CD45$^-$ cells were combined in a 1:1 ratio at the final concentration of 1000 cells/µl in PBS-BSA for scRNA-seq library preparation.

## Next generation sequencing

**RNA-seq.** End-stage macro-dissected HCC nodules, control livers, and cell lines non-treated or treated with Vx-11e were processed for RNA-seq as follows. Tissue samples were homogenized in 600 µl of buffer RLT (79216, Qiagen) with the addition of 1% B-mercaptoethanol using the TissueLyserII (85300, Qiagen) in combination with 5 mm stainless steel beads (69989, Qiagen). Cells were lysed with 350ul of buffer RLT (79216, Qiagen) and the total RNA was isolated using the RNeasy Mini Kit (74106, Qiagen), according to the manufacturer's instructions. The library preparation for bulk tumors and cell lines was generated using the TruSeq Stranded mRNA sample preparation kit (Illumina Inc., San Diego, RS-122-2101/2) according to the manufacturer's instructions (Illumina, Document #1000000040498 v00). The libraries were sequenced with 54 paired-end reads on a NovaSeq-6000 using a Reagent Kit v1.5 (Illumina Inc., San Diego). For CD45[+] RNA-seq and the cell line RNA-seq, the libraries were generated using Smart-Seq2 RNA (Illumina) according to the manufacturer's instructions and paired-end sequenced using the Nextseq-550 High Output Kit v2.5.

## Whole exome sequencing (WES)

For WES, macro-dissected HCC were weighted and equal amounts (-1 mg each) from end-stage ($n = 3$ for *Myc*[OE]/*Trp53*[KO], *Myc*[OE]/*Pten*[KO], and *Nras*[G12D]/*Pten*[KO], $n = 3$ for *Nras*[G12V]/*Pten*[KO] and $n = 1$ for pT3-*Nras*[G12V]/*Pten*[KO]) HCC-bearing mice and control liver ($n = 3$). Following DNA extraction and fragmentation (200–300 bp), a maximum of 1 µg of sheared DNA was used for library preparation using the KAPA HTP Prep Kit (KAPA Biosystems, KK8234) following manufacturer's instructions. The samples were 100 bp paired-end sequenced Illumina Novaseq 6000.

## scRNA-seq

Chromium Controller platform of 10X Genomics was used for single-cell partitioning and barcoding. Single Cell 3′ Gene Expression were prepared according to the manufacturer's protocol "Chromium NextGEM Single Cell 3′ Reagent Kits v3.1" (CG000315, 10X Genomics). A NovaSeq 6000 Illumina sequencing system was used for paired end sequencing of the Single Cell 3′ Gene Expression libraries at a sequencing depth of -17.000 mean reads per cell for control liver and 35.000 mean reads per cell for *Nras*[G12D]/*Pten*[KO]. NovaSeq 6000 paired end sequencing was performed using NovaSeq SP Reagent Kit v1.5 (cat# 20028401, Illumina) and NovaSeq S1 Reagent Kit v1.5 (cat# 20028319, Illumina).

## Bioinformatic analyses

All the analyses were performed using R (4.2.2, 2022-10-31), unless mentioned otherwise. Plots were generated using the package *ggplot2* (v.3.4.2) or *ggpubr* (v.0.6.0), except where mentioned.

## RNA-seq

Bulk tumors, CD45[+] and cell lines RNA-seq were analyzed after paired-end sequencing reads were trimmed with *seqpurge* (v. 2022_11) and aligned with *Hisat2* (v.2.1.0) using as reference GRCm38 (mm10) (--min-intronlen 20 --max-intronlen 500000 -k 5 --minins 0 --maxins 500 --fr --new-summary --threads 16). The read counts were generated using the function *gensum* (v.0.2.1) with gtf (v. 100) annotation. PCA was performed to verify accordance between replicates. The differential expression analysis was performed using *EdgeR*(v.3.40.2)[115]. The samples were trimmed-mean-of-M-values (TMM) normalized to account for library size. To verify replicate agreement, the Pearson correlation was calculated on the TMM-normalized log$_2$(TPMs) of each replicate. To test for differential expression, the generalized linear model method using the *glmLRT* function was used with a significance cutoff of FDR 1%. The log$_2$(TPMs) were also calculated using the normalized read counts. The signatures for each genetically-distinct HCC bulk tumor samples were defined as the DEG between each model relative to the HDTVi control liver.

## Classification of murine HCC subtypes according to human molecular HCC subtypes

For the prediction of classification, the algorithm *NearestTemplatePrediction* (v.4 2015-12-02) was used on the Gene Pattern public server[116]. The input files of each of the replicates from the RNA-seq of the bulk tumor HCC models or CD45[+] cells were TMM-normalized TPMs in a GCT format, the classes of the different classifications were based on published references[63–68,75,103]. Statistical significance cutoff was defined as $p < 0.01$ with multiple test correction using the Bonferroni method.

## Human TCGA:LIHC and pan-cancer data analysis

TCGA pan-cancer and LIHC data were obtained from http://gdac.broadinstitute.org/ using the package *UCSCXenaTools* (v.1.4.8)[117]. The raw counts were processed using *EdgeR* (v.3.40.2)[115] to obtain TMM-normalized log$_2$(TPMs). Only TCGA:LIHC patients with clinical and expression data were considered for downstream analysis ($n = 372$). The segregation of patients by high/low correlation was determined by calculating the Pearson correlation between the signature of the genetically-distinct HCC models and the TCGA: LIHC patients using their log$_2$(TPMs). To define the patients with high correlation, the cutoff was a correlation within the 4th quantile of the data ($n = 93$) (top 25%), whereas the low correlation patients were selected based on a correlation within the the 1st quantile ($n = 93$) (bottom 25%). The conversion of mouse gene symbols to human was based on homology using ensembl[118]. For the calculation of the survival curves, the Kaplan–Meier estimator was obtained with the *survfit* function from the package "survival" (v.3.4-0, https://github.com/therneau/survival). The statistical test applied was the log-rank test using the 'survminer' package (v.0.4.9, https://github.com/kassambara/survminer). The correlation between the TCGA: LIHC patients per iCluster and the models was performed on the log$_2$(TPMs) of all genes. The highest Pearson correlation was used as the determinant value to define iCluster for each murine model. Significance was tested on the Pearson's coefficient. The function *Heatmap* from the library "ComplexHeatmap" (v.2.14.0) was used to plot the heatmaps with the gene expression per patient. The gene expression is z-scored normalized per gene.

## Enrichment analysis

The enrichment of MAPK (KEGG_MAPK_SIGNALING_PATHWAY), PIK3 (REACTOME_PI3K_AKT_SIGNALING_IN_CANCER) and MYC (HALLMARK_MYC_TARGETS_V1) signaling pathways in the bulk tumor RNA-seq was performed using the *HypeR* (v.1.14.0) package[119] with an hypergeometric test and for TCGA patients, we applied single sample gene set enrichment analysis (ssGSEA v.10.0.11)[120] The gene set enrichment analysis in the CD45[+] RNA-seq was performed with GSEA (v.4.2.2) using immune genesets[76] to define the pan-immune categories with default parameters (permutation type: gene_set; number of permutations: 1000). To determine the enrichment of Biocarta and KEGG pathways, we applied single sample gene set enrichment analysis (ssGSEA v. 10.0.11)[120] on the Gene Pattern public server[116]. The enrichment of GM-CSF signature (*Csf2rb*[KO] - WT[89]); was performed using the *Hyper* (v.1.14.0) package[119] with an hypergeometric test. The gene set enrichment analysis of DEG between *Nras*[G12D] relative to *Nras*[G12V] cancer cells and bulk tumor was determined using enrichR (v.3.2) and MSigDB Hallmarks database.

## Protein-protein interaction analysis

The protein-protein interaction analysis was performed using Metascape(v.3.5.20240101)[121]. In short, a network is created using

databases for protein interactions. The resulting network is then tested for pathway enrichment.

## scRNA-seq analyses

The sequencing reads were mapped using Cell Ranger (v.6.1.2) to genome build GRCm38 (mm10). All downstream processing and analysis were performed using the R package *Seurat* (v.4.1.3). The counts were loaded into R and to remove low quality cells, only features (genes) detected in a minimum of three cells and cells expressing a minimum of 200 features (genes) were considered for the further quality control. To further filter out low quality cells with high percentage of mitochondrial reads and aberrantly low/high number of gene expressed depending on each sample, we used the median absolute deviation higher (for mitochondrial percentage) or lower (for the number of genes detected) than the median value for these metrics to remove cells. A total of 12,313 cells remained and the samples were SCT normalized and integrated using the *IntegrateData()* function. The data was reduced and clustered (Louvain algorithm, 0.075 resolution) which were then annotated as Neutrophils, Monocytic cells, cDC1s, pDCs, B cell, T cell, and NK cells (see Supplementary Data 7). The differentially expressed genes were calculated with the function *FindAllMarkers*. Re-clustering of the monocytic cells and the cDC1 using 0.25 resolution. The differential abundance was determined using the *MiloR* (v.1.2.0) package (https://marionilab.github.io/miloR). Briefly, this method is based on assigning cells to 'neighborhoods' from a k-nearest neighbor graph; the counts are then fitted using a negative binomial General Linear Model. This model is then used to test for differential abundance of cell proportions in the "neighborhoods". The functions *enricher* and *compareCluster* from the R package cluster-Profiler (v.4.6.2) were used to calculate the enrichment of the up-regulated DEG between *Nras*[G12D]/*Pten*[KO] relative to control using the MSigDB Hallmarks database.

To integrate the annotation from both human HCC and our murine scRNA-seq, the function *FindTransferAnchors()* and *MapQuery()* were used from *Seurat* (v.4.1.3). The predicted annotations from ref. 75 were projected on our UMAP to confirm correct annotation of the cell types.

## Whole exome sequencing analysis

The paired-end reads were trimmed using SeqPurge and aligned with Burrows–Wheeler alignment (v.0.5.10) to GRCm38 (mm10). PCR duplicates were designated using rumidup (https://github.com/NKI-GCF/rumidup). For basecall recalibration and variant calling, the BaseRecalibrator and Mutect2 functions from the genome analysis toolkit[122] (GATK; v.4.2.6.0) and were applied, respectively. SNPEff[123] and SnpSift[124], (v.5.1d), were used to annotate and select the somatic variants calls with TLOD greater or equal than 15 and filtered on the disruptive effects (conservative inframe deletion, disruptive inframe deletion, disruptive inframe insertion, frameshift variant, missense variant, start lost, stop gained, stop lost or stop retained variant). The counts of variants generated by SNPEff were used to calculate the TMB as follows: n * 1,000,000/covered_bases, where the covered_bases were the number of bases that had at least 5x coverage, which varied between 37.47 M and 37.48 M bases per sample.

The TMB from human LIHC was obtained from https://www.cbioportal.org/ from different datasets[125–127], and the simple nucleotide variation data was downloaded with TCGAbiolinks[128] (v. 2.22.4). The variant types and base substitutions were extracted using maftools[129] (v. 2.10.05). Analyses were done using the statistical programming language R version 4.0.3.

## Transcription factor motif analyses

The genes of interest were defined as the DEG between *Nras*[G12D]/*Pten*[KO] cells and *Nras*[G12V]/*Pten*[KO] that overlapped changes in Vx-11 treated compared to control in *Nras*[G12D]/*Pten*[KO] cells and that excluded DEGs of *Nras*[G12V]/*Pten*[KO] cells also treated with Vx-11 (see Supplementary Data 9). The genes selected for motif enrichment were both up-regulated in *Nras*[G12D]/*Pten*[KO] in Vx-11e cells and compared to *Nras*[G12V]/*Pten*[KO] cells. The motif enrichment was performed in the promoter regions extracted with *get_biomart_promoters* function from PromoterOntology (v.0.0.1) of mm10. The Simple Enrichment Analysis (SEA v. 5.4.1) tool from the MEME suite[130] (sea --verbosity 1 --oc. --thresh 10.0 --align center –p file.fa --m db/MOUSE/HOCOMOCOv11_full_MOUSE_mono_meme_format.meme).

## Statistics and reproducibility

For boxplots, the boxes represent data from the 1st to the 3rd quartile, the middle line represents the median, the whiskers denote a minimum (1st quartile – 1.5*(1st −3rd quartile)) and a maximum (3rd quartile +1.5*(1st −3rd quartile)), and individual points indicate the outliers.

The statistical significance from differential gene expression analysis by EdgeR was determined by a two-sided negative binomial generalized linear model likelihood ratio test and followed by a Benjamini-Hochberg multiple testing correction.

## Reporting summary

Further information on research design is available in the Nature Portfolio Reporting Summary linked to this article.

## Data availability

The RNA-seq, scRNA-seq and WES publicly available data generated in this study are available in the Gene Expression Omnibus (GEO) database under accession code GSE216717. The GM-CSF signature was derived from PRJEB9884. The human HCC data was sourced from PRJCA007744. The TCGA publicly available data used in this study are available in https://portal.gdc.cancer.gov. The remaining data are available within the Article, Supplementary Information or Source Data file. Source data are provided with this paper.

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

## Acknowledgements

We thank Drs L. Zender and S. Lowe for the kind gift of the *Ras* plasmids, U. Hibner for providing the AML12 cell line, I. Michael for providing the B20S VEGF neutralizing antibody, and K.E de Visser for sharing the HEK 293 T cells. We thank all the members of the Akkari lab for insightful comments and discussion, and Jules Gadiot, Tamara Filipovic, Martina Färber, Naz Kocabay, and Cankat Ertekin for excellent technical support. We are grateful to the facilities of the Netherlands Cancer Institute: Flow Cytometry, Animal Laboratory, Mouse Clinic Imaging Unit and Genomics, and to Dr Ji-Ying Song from the Experimental Animal Pathology for her help with the histopathological evaluation. We thank members of the Karin de Visser and René Bernards laboratories for insightful discussion during the preparation of the manuscript and for sharing reagents. This research was supported by the Dutch Cancer Society (KWF 12049 to L.A. and KWF 13476 to S.V.), Oncode Institute (L.A.), Cancer Genomics Center (L.A.) and of the Dutch Ministry of Health, Welfare and Sport. The results presented (Figs. 2b, c, 5i–k, 6h, Supplementary Figs. 2b–f, 5o, p) relies partially on data produced by the TCGA Research Network at https://www.cancer.gov/tcga. Figures 1a, 7d, Supplementary Figs. 7a, 8a, 9i were created with BioRender.com.

## Author contributions

L.A., S.V., C.F.A.R., and D.T. conceived the study, designed experiments, interpreted data. L.A., S.V., D.T., C.F.A.R, and M.A.K. wrote the manuscript. C.F.A.R., D.T., S.V., M.H.P.G., and E.T. performed and analyzed experiments. M.A.K. performed all computational analyses. D.J.K. contributed to the CD45⁺ RNAseq analysis. D.G. and R.J.C.K. performed the WES analysis. Z.H. and J.X. performed and analyzed the HCC patient TMA. J.V. performed histological analyses. All authors edited or commented on the manuscript.

## Competing interests

The authors declare no competing interests.

## Additional information

[1]Division of Tumor Biology and Immunology, The Netherlands Cancer Institute, Amsterdam, The Netherlands. [2]Oncode Institute, The Netherlands Cancer Institute, Amsterdam, The Netherlands. [3]State Key Laboratory of Oncology in South China, Sun Yat-sen University Cancer Center, Guangzhou 510060, PR China. [4]Genomics Core facility, The Netherlands Cancer Institute, Amsterdam, The Netherlands. [5]Department of Pathology, Cancer Center Amsterdam, Amsterdam UMC, University of Amsterdam, Amsterdam, the Netherlands. [6]These authors contributed equally: Christel F. A. Ramirez, Daniel Taranto, Masami Ando-Kuri. [7]These authors jointly supervised this work: Serena Vegna, Leila Akkari. ✉e-mail: s.vegna@nki.nl; l.akkari@nki.nl

