## [Peer Review File · Nature Communications]

Cancer cell genetics shaping of the tumor microenvironment reveals myeloid cell-centric exploitable vulnerabilities in hepatocellular carcinomaEditorial Note: This manuscript has been previously reviewed at another journal that is not operating a transparent peer review scheme. This document only contains reviewer comments and rebuttal letters for versions considered at *Nature Communications*.

REVIEWER COMMENTS

Reviewer #4 (Remarks to the Author):

Major comments:

1. The authors analyzed the correlation between MycOE- and NrasG12D-driven murine HCCs and the human HCC TMA dataset. The results would be more relevant if they incorporated more human HCC samples for pERK1/2 stratification analysis.
2. In the analysis of the TCGA-LIHC dataset, the authors set a threshold where patients with high correlation of signatures were those above the 3rd quantile, and those with low correlation were below the 1st quantile. Why? Also, in Fig. 5I, it is unclear why there are 66 and 65 samples for NrasG12D/PtenKO and NrasG12V/PtenKO signatures, respectively, instead of 93, as mentioned in the methods section. Please explain the discrepancy in the number of samples used.
3. Regarding the mechanism of benefit when combining anti-VEGF and anti-GM-CSF therapies, the authors indicated that VEGF blockade enhances cancer cell necrotic death orthogonal to anti-GM-CSF-induced apoptosis. This increase is mediated by inhibiting the VEGF signaling pathway in cancer cells. The authors should present more evidence for this mechanism in cancer cells.
4. In Fig. 8B, the combination of a-GM-CSF with a-VEGF shows promising efficacy. Is it correct that the a-GM-CSF+a-PD-L1 had only 4 mice? Also, given that the standard of care for HCC is dual inhibition of VEGF and PD-L1 and that anti-VEGF therapy alone is not efficacious, investigating the combination of a-GM-CSF with this dual inhibition regimen could have been more relevant. Adding some rationale for this approach in the discussion would enhance the relevance of the findings.

Minor comments:

- In Fig. 3C The bars indicating statistical significance should be clearer with respect to which

groups are compared.

- The labels of interferon signaling in Fig. S3I are overlapping.
- Figures S4H and S4G are mistakenly switched in the manuscript.
- Fig. S5K, line 395: To demonstrate that GM-CSF expression was specific to the tumor, normal liver tissue from tumor-bearing mice should also be analyzed.
- Fig. 8C: quantification of cleaved caspase 3 by surface area is unusual. Normalizing area per number of cells might be more accurate.

Reviewer #4 (Remarks to the Author):

We thank the reviewer for her/his insightful comments. We believe that the changes made in the manuscript in response to the reviewer's suggestions have enhanced the clarity of the results and the methodology.

Major comments:

1. The authors analyzed the correlation between MycOE- and NrasG12D-driven murine HCCs and the human HCC TMA dataset. The results would be more relevant if they incorporated more human HCC samples for pERK1/2 stratification analysis.

In the first version of our manuscript and all versions thereafter, we presented extensive pathway analyses across the TCGA-LIHC dataset and validated the positive association between MAPK enrichment and patients' correlation to both *Nras*-driven signatures. Importantly, *Nras*^{G12D}/*Pten*^{KO}-correlating patients exhibited significantly higher MAPK activation compared to their *Nras*^{G12V}/*Pten*^{KO} counterparts (**Fig. 2B** and **S2C** in revised manuscript). Orthogonally, in additional analyses performed following the previous review round, we have stratified TCGA-LIHC patients according to their MAPK pathway enrichment score and correlated the highly enriched MAPK patients to the *Nras*-driven murine HCC signatures. Our results indicate that patients with increased MAPK activation significantly resemble the two *Nras*-driven models (**Rebuttal Fig. 1**). Importantly, patients presenting high correlation with *Nras*^{G12D}/*Pten*^{KO} murine HCC showed overall heightened induction of the MAPK-signalling pathway, including MAPK1 and MAPK3 (ERK1 and ERK2) but not only, compared to the *Nras*^{G12V}/*Pten*^{KO} high correlation patients (new **Fig. S2D** in revised manuscript; lines 194-200). Additionally, we reported the correlation between the histopathological and prognostic features of the *Nras*^{G12D} and *Nras*^{G12V} murine models and the patients whose transcriptomic profile closely resembled the liver models they related to (**Fig. 2A** in revised manuscript). Importantly, we have evaluated the resemblance of the genetically-distinct murine models to the recently published liver tumor immune microenvironment (TIME) subtypes¹ and revealed a significant correlation between the *Nras*^{G12V}/*Pten*^{KO} TME with the TIME-IA (Immune Active) subtype (**Fig. 3E** in revised manuscript), associated with significant T cell activation and infiltration, in line with our findings (**Fig. 3C** and **S3B** in revised manuscript). Distinctively, *Nras*^{G12D}/*Pten*^{KO} and *Myc*^{OE} TMEs significantly correlated with the TIME-ISM (Immunosuppressive Myeloid) subtype (**Fig. 3E** in revised manuscript), associated with poor patient prognosis, myeloid cell dominance, and increased expression of both immunosuppressive and IL1-related inflammatory signalling pathways. Altogether, we believe this analysis fulfilled the correlation of distinct cancer cell-intrinsic oncogenic signalling pathways, more particularly in terms of MAPK intensity status, to patient clinical features to a broader extent than pERK1/2 stratification status alone.

We further addressed this point following the reviewer's request to expand the translational relevance of our work. We did so by extending and further validating the correlation between the activation of distinct tumor cell intrinsic-oncogenic signalling pathways and the specific TME reshaping reflecting patient prognosis in a large dataset of HCC sample TMA (**Fig 2D-E, 3D** and **S3G** in revised manuscript). As explained in our previous answer and reiterated in the manuscript (lines 207-211), this collection of clinical samples primarily consisted of poorly to moderately-differentiated HCCs characterized by hepatitis B virus infection, closely resembling *Nras*^{G12D}/*Pten*^{KO}- and *Myc*^{OE}-induced

murine HCC models. Notably, this dataset does not contain NASH-like patient samples, which would more closely resemble a well-differentiated, possibly pERK1/2 intermediately-positive subset. Nonetheless, within this dataset of aggressive HCCs, we were able to observe a correlation between distinct cancer cell-intrinsic oncogenic signalling pathways and TME composition and phenotype and patient prognosis. Thus, we believe this added information significantly strengthened our manuscript, where we focused mostly on the intricacies of the aggressive *Nras*^{G12D}/*Pten*^{KO} HCC model.

We have carefully considered the practical implications of incorporating additional human samples for the analysis of pERK1/2 stratification. Acquiring and characterizing these samples would require a substantial investment of time and resources. It is crucial to note that the prevailing epidemiological and etiological factors associated with HCC in China (where the dataset under consideration has been collected and analysed) indicate that NASH is not yet endemic/widespread and the majority of cases are still related to hepatitis infections², thereby posing challenges in acquiring a sufficient number of samples of this type. Therefore, it appears to be unrealistic/impractical to obtain a significant representation of the NASH-like HCCs - corresponding to the *Nras*^{G12V}/*Pten*^{KO} model - from the same data source. Considering these limitations, we firmly believe that this is beyond the scope of the paper and we are confident the data we present in the current version of the manuscript represents satisfactory evidence for the role of cancer cell-intrinsic oncogenic signalling pathways in shaping HCC multi-layered heterogeneity.

Figure 1. Barplots depicting the number of patients with high/low enrichment of MAPK signaling pathway while simultaneously presenting high (red)/low (grey) correlation with *Nras*^{G12D}/*Pten*^{KO} (left) or *Nras*^{G12V}/*Pten*^{KO} (right) signature. Patients were divided into high/low according to median of enrichment score of MAPK.

2. In the analysis of the TCGA-LIHC dataset, the authors set a threshold where patients with high correlation of signatures were those above the 3rd quantile, and those with low correlation were below the 1st quantile. Why? Also, in Fig. 5I, it is unclear why there are 66 and 65 samples for *Nras*^{G12D}/*Pten*^{KO} and *Nras*^{G12V}/*Pten*^{KO} signatures, respectively, instead of 93, as mentioned in the methods section. Please explain the discrepancy in the number of samples used.

We believe this might have been a misunderstanding in the way we laid out the definition of the quantiles we set. The entire TCGA dataset (372 LIHC patients) has been divided in four quantiles, each one consisting of 93 patients. For these analyses, we considered the most extreme scenarios by arbitrarily designating patients with low correlation as those within the 1st quantile (bottom 25%), and patients with high correlation as those above the 3rd quantile, thus corresponding to the 4th quantile (top 25%). To address this clarification, we have appropriately modified the Methods section in the revised manuscript accordingly (lines 1013-1016).

Regarding the second point of the Reviewer, the correlation score between the transcriptome profile of TCGA patients and each of the murine HCC models gives rise to overlapping patients assigned to high and low correlation groups, as shown in **Fig. S2D** in the revised manuscript. Hence, to directly compare patients assigned to high correlation with *Nras*^{G12D} and *Nras*^{G12V} signatures, we excluded the overlapping individuals and used the patients unique to each group, 65 and 66 patients respectively for *Nras*^{G12D}/*Pten*^{KO} for *Nras*^{G12V}/*Pten*^{KO} as shown in **Fig. 5I** in revised manuscript. We have now clarified this point in the legend.

3. Regarding the mechanism of benefit when combining anti-VEGF and anti-GM-CSF therapies, the authors indicated that VEGF blockade enhances cancer cell necrotic death orthogonal to anti-GM-CSF-induced apoptosis. This increase is mediated by inhibiting the VEGF signaling pathway in cancer cells. The authors should present more evidence for this mechanism in cancer cells.

In response to the reviewer's concern, we treated *ex vivo* *Nras*^{G12D}/*Pten*^{KO} HCC cell line with either a-IgG, a-GM-CSF (5 µg/mL), a-VEGF (75 or 150 µg/mL), or combined a-GM-CSF (5ug/mL) + a-VEGF (75 or 150 ug/mL) and assessed by flow cytometry the level of RIP3, a kinase used as a marker to identify cells undergoing necroptosis³. This approach was chosen because, to our knowledge, there is currently no reliable marker available to specifically assess necrosis. Interestingly, we observed that VEGF blockade led to a significant increase in RIP3⁺ cells across the two tested concentrations in this setting (**Rebuttal Fig. 2**), suggesting that curbing VEGF signalling can promote necroptosis of *Nras*^{G12D}/*Pten*^{KO} HCC cells in a cancer cell-intrinsic manner, likely due to autocrine and/or paracrine pro-survival signalling mediated by cancer cell-induced VEGF secretion. Conversely, a-GM-CSF treatment exhibited no impact on RIP3 expression in cancer cells as monotherapy or in combination with a-VEGF (**Rebuttal Fig. 2**). This observation is consistent with the cancer cell-extrinsic effect of GM-CSF observed in the TME, as reported in the manuscript (**Fig. 7E-H, S7E,H and S8I-L** in revised manuscript). Overall, our findings indicate that necroptosis induced by VEGF blockade synergizes with a-GM-CSF-induced cancer cell apoptosis mediated by the TME. However, as highlighted in the manuscript, VEGF expression is not restricted to the cancer cells, but it is also highly expressed in the *Nras*^{G12D}/*Pten*^{KO} HCC TME (**Fig. 4C, F and 5C** in the revised manuscript), thereby suggesting that the *ex vivo* system presented here may not fully recapitulate the effect observed *in vivo*. Furthermore, it is important to mention that the tissue analysis for necrosis shown in **Fig. 8E** does not enable to discriminate between different forms of programmed and non-programmed cell death. Therefore, it is possible that the results described in the *in vivo* setting (**Fig. 8E**) might arise from a combination of several cell death processes. Therefore, in the manuscript, we specifically refer to the *in vivo* models when discussing the impact of dual VEGF and GM-CSF blockade in *Nras*^{G12D}/*Pten*^{KO} HCC-bearing mice.

Figure 2

Figure 2. Barplot depicting the percentage of RIP3⁺ *Nras*^{G12D}/*Pten*^{KO} cells relative to total live cells treated with either anti-IgG (anti-IgG2a+anti-IgG1), anti-GM-CSF, anti-VEGF or the combination at the concentrations as indicated for 120 hours. N=3 performed in triplicate. Graph shows mean ± SEM. Statistical significance was determined by unpaired student T-test versus anti-IgG. ***p < 0.001, ****p < 0.0001

4. In Fig. 8B, the combination of a-GM-CSF with a-VEGF shows promising efficacy. Is it correct that the a-GM-CSF+a-PD-L1 had only 4 mice? Also, given that the standard of care for HCC is dual inhibition of VEGF and PD-L1 and that anti-VEGF therapy alone is not efficacious, investigating the combination of a-GM-CSF with this dual inhibition regimen could have been more relevant. Adding some rationale for this approach in the discussion would enhance the relevance of the findings.

As outlined in our previous answer, the number of animals used in the a-GM-CSF + a-PD-L1 arm (n=4) was sufficient to show the increased survival benefit of the combination arm compared to a-GM-CSF alone. Adhering to the principle of the 3R (replacement, reduction and refinement), we reasoned that enrolling additional mice in this trial would be unnecessary to reaffirm this conclusion.

As rightfully pointed out by the Reviewer, the a-GM-CSF + a-VEGF combination strategy significantly prolonged the survival of *Nras*^{G12D}/*Pten*^{KO} HCC-bearing mice compared to the other models (Fig. 8B in revised manuscript). Regarding the suggestion to enrol mice in a triple combination trial (a-GM-CSF + a-VEGF + a-PD-L1), we initially opted against pursuing this treatment strategy due to concerns about a potential toxicity associated with administering three drugs simultaneously. We have incorporated additional sentences in the Results section (lines 564-565) and in the Discussion section (lines 639-645) to provide a more comprehensive explanation of this decision.

Concerning the Reviewer's remark about the presumed lack of efficacy of VEGF blockade alone, there are noteworthy reports suggesting the effectiveness of bevacizumab as a standalone treatment in HCC patients. For instance, a Phase II clinical trial demonstrated an improvement in the progression-free survival rate comparable to sorafenib⁴. Additionally, some patients achieved a partial response in another Phase II trial⁵. These findings imply that a specific subset of HCC patients might exhibit heightened responsiveness to VEGF blockade, and would potentially benefit from combinatorial approaches with GM-CSF blockade, a point we further highlighted in our manuscript conclusion (lines 646-652).

Minor comments:

- In Fig. 3C The bars indicating statistical significance should be clearer with respect to which groups are compared.

We would like to emphasize that, as appropriately illustrated in **Fig. S3C** in the revised manuscript, all statistical analyses were performed within each HCC mouse model, and not between them. In other words, statistical comparisons were performed between each tumor model at different stages of disease progression and its relative controls, but not between genetically distinct HCC tumor samples. To further clarify this point, we have included a sentence that explains how the groups were analysed in the legend of **Fig. S3C** (lines 1209-1212 of the revised manuscript).

- The labels of interferon signaling in Fig. S3I are overlapping.

As requested by the Reviewer, we have relocated the labels in **Fig. S3I** in the revised manuscript to a more prominent position, thus limiting text overlap and enhancing readability.

- Figures S4H and S4G are mistakenly switched in the manuscript.

We thank the reviewer for her/his remark. Indeed, the order of Figures S4H and S4G was incorrect. We are now presenting the results in the correct order and edited the references in the manuscript accordingly (lines 352 and 355).

- Fig. S5K, line 395: To demonstrate that GM-CSF expression was specific to the tumor, normal liver tissue from tumor-bearing mice should also be analyzed.

We would like to clarify that Figure S5K refers to the cytokine array performed on the sera of tumor-bearing mice and control empty vector-injected HDTV mice. In this context, we have underscored that the release of GM-CSF in the *Nras*^{G12D}/*Pten*^{KO} mouse model was specific to cancer cells from this genotype, local to the tumor region (lines 396-398) and not systemically observed (**Fig. S5D** and **S5C,K** in revised manuscript). Importantly, this increase was also observed when compared to the non-transformed hepatocyte cell line (**Fig. S5D** in revised manuscript). Altogether, we have provided significant evidence that *Nras*^{G12D}/*Pten*^{KO} cancer cells represent the major source of GM-CSF at the tumor bulk level. The unavailability of peritumoral regions from these animals prevents us from assessing the level of GM-CSF at the peritumoral bulk level and, considering the points mentioned earlier, we propose that such assessment will not affect our conclusions.

- Fig. 8C: quantification of cleaved caspase 3 by surface area is unusual. Normalizing area per number of cells might be more accurate.

We would like to clarify that **Fig. 8C** in the manuscript was quantified by assessing the number of positive cells per nodule area, not by positive surface area. We have therefore corrected the Methods section (line 905) in the revised manuscript and changed the legend in **Fig. 8D** (line 1373) accordingly.

References

- 1 Xue, R. *et al.* Liver tumour immune microenvironment subtypes and neutrophil heterogeneity. *Nature* **612**, 141-147 (2022). <https://doi.org:10.1038/s41586-022-05400-x>
- 2 Lin, J. *et al.* Epidemiological Characteristics of Primary Liver Cancer in Mainland China From 2003 to 2020: A Representative Multicenter Study. *Front Oncol* **12**, 906778 (2022). <https://doi.org:10.3389/fonc.2022.906778>
- 3 Moriwaki, K. & Chan, F. K. RIP3: a molecular switch for necrosis and inflammation. *Genes Dev* **27**, 1640-1649 (2013). <https://doi.org:10.1101/gad.223321.113>
- 4 Boige, V. *et al.* Efficacy, safety, and biomarkers of single-agent bevacizumab therapy in patients with advanced hepatocellular carcinoma. *Oncologist* **17**, 1063-1072 (2012). <https://doi.org:10.1634/theoncologist.2011-0465>
- 5 Ren, Z. *et al.* Sintilimab plus a bevacizumab biosimilar (IBI305) versus sorafenib in unresectable hepatocellular carcinoma (ORIENT-32): a randomised, open-label, phase 2-3 study. *Lancet Oncol* **22**, 977-990 (2021). [https://doi.org:10.1016/S1470-2045\(21\)00252-7](https://doi.org:10.1016/S1470-2045(21)00252-7)

REVIEWERS' COMMENTS

Reviewer #4 (Remarks to the Author):

The changes and responses are satisfactory. I have no other major comments.